# CLIPSep: Learning Text-queried Sound Separation with Noisy Unlabeled Videos

**Hao-Wen Dong**[1,2*]   **Naoya Takahashi**[1†]   **Yuki Mitsufuji**[1]
**Julian McAuley**[2]   **Taylor Berg-Kirkpatrick**[2]

[1]Sony Group Corporation   [2]University of California San Diego
hwdong@ucsd.edu, {Naoya.Takahashi,Yuhki.Mitsufuji}@sony.com, {jmcauley,tberg}@ucsd.edu

## Abstract

Recent years have seen progress beyond domain-specific sound separation for speech or music towards universal sound separation for arbitrary sounds. Prior work on universal sound separation has investigated separating a target sound out of an audio mixture given a text query. Such text-queried sound separation systems provide a natural and scalable interface for specifying arbitrary target sounds. However, supervised text-queried sound separation systems require costly labeled audio-text pairs for training. Moreover, the audio provided in existing datasets is often recorded in a controlled environment, causing a considerable generalization gap to noisy audio in the wild. In this work, we aim to approach text-queried universal sound separation by using only unlabeled data. We propose to leverage the visual modality as a bridge to learn the desired audio-textual correspondence. The proposed CLIPSep model first encodes the input query into a query vector using the contrastive language-image pretraining (CLIP) model, and the query vector is then used to condition an audio separation model to separate out the target sound. While the model is trained on image-audio pairs extracted from unlabeled videos, at test time we can instead query the model with text inputs in a zero-shot setting, thanks to the joint language-image embedding learned by the CLIP model. Further, videos in the wild often contain off-screen sounds and background noise that may hinder the model from learning the desired audio-textual correspondence. To address this problem, we further propose an approach called *noise invariant training* for training a query-based sound separation model on noisy data. Experimental results show that the proposed models successfully learn text-queried universal sound separation using only noisy unlabeled videos, even achieving competitive performance against a supervised model in some settings.

## 1 Introduction

Humans can focus on to a specific sound in the environment and describe it using language. Such abilities are learned using multiple modalities—auditory for selective listening, vision for learning the concepts of sounding objects, and language for describing the objects or scenes for communication. In machine listening, selective listening is often cast as the problem of sound separation, which aims to separate sound sources from an audio mixture (Cherry, 1953; Bach & Jordan, 2005). While text queries offer a natural interface for humans to specify the target sound to separate from a mixture (Liu et al., 2022; Kilgour et al., 2022), training a text-queried sound separation model in a supervised manner requires labeled audio-text paired data of single-source recordings of a vast number of sound types, which can be costly to acquire. Moreover, such isolated sounds are often recorded in controlled environments and have a considerable domain gap to recordings in the wild, which usually contain arbitrary noise and reverberations. In contrast, humans often leverage the visual modality to assist learning the sounds of various objects (Baillargeon, 2002). For instance, by observing a dog barking, a human can associate the sound with the dog, and can separately learn that the animal is called a "*dog.*" Further, such learning is possible even if the sound is observed in a noisy environment, e.g.,

---

*Work done during an internship at Sony Group Corporation

†Corresponding author

when a car is passing by or someone is talking nearby, where humans can still associate the barking sound solely with the dog. Prior work in psychophysics also suggests the intertwined cognition of vision and hearing (Sekuler et al., 1997; Shimojo & Shams, 2001; Rahne et al., 2007).

Motivated by this observation, we aim to tackle text-queried sound separation using only unlabeled videos in the wild. We propose a text-queried sound separation model called CLIPSep that leverages abundant unlabeled video data resources by utilizing the contrastive image-language pretraining (CLIP) (Radford et al., 2021) model to bridge the audio and text modalities. As illustrated in Figure 1, during training, the image feature extracted from a video frame by the CLIP-image encoder is used to condition a sound separation model, and the model is trained to separate the sound that corresponds to the image query in a self-supervised setting. Thanks to the properties of the CLIP model, which projects corresponding text and images to close embeddings, at test time we instead use the text feature obtained by the CLIP-text encoder from a text query in a zero-shot setting.

However, such zero-shot modality transfer can be challenging when we use videos in the wild for training as they often contain off-screen sounds and voice overs that can lead to undesired audio-visual associations. To address this problem, we propose the *noise invariant training* (NIT), where query-based separation heads and permutation invariant separation heads jointly estimate the noisy target sounds. We validate in our experiments that the proposed noise invariant training reduces the zero-shot modality transfer gap when the model is trained on a noisy dataset, sometimes achieving competitive results against a fully supervised text-queried sound separation system.

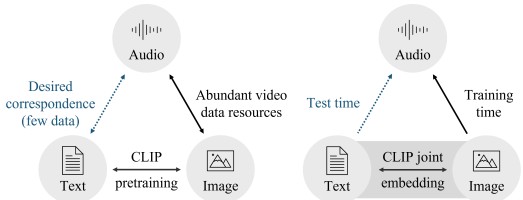

Figure 1: An illustration of modality transfer.

Our contributions can be summarized as follows: 1) We propose the first text-queried universal sound separation model that can be trained on unlabeled videos. 2) We propose a new approach called *noise invariant training* for training a query-based sound separation model on noisy data in the wild. Audio samples can be found on an our demo website.[1] For reproducibility, all source code, hyperparameters and pretrained models are available at: `https://github.com/sony/CLIPSep`.

## 2 RELATED WORK

**Universal sound separation**   Much prior work on sound separation focuses on separating sounds for a specific domain such as speech (Wang & Chen, 2018) or music (Takahashi & Mitsufuji, 2021; Mitsufuji et al., 2021). Recent advances in domain specific sound separation lead several attempts to generalize to arbitrary sound classes. Kavalerov et al. (2019) reported successful results on separating arbitrary sounds with a fixed number of sources by adopting the *permutation invariant training* (PIT) (Yu et al., 2017), which was originally proposed for speech separation. While this approach does not require labeled data for training, a post-selection process is required as we cannot not tell what sounds are included in each separated result. Follow-up work (Ochiai et al., 2020; Kong et al., 2020) addressed this issue by conditioning the separation model with a class label to specify the target sound in a supervised setting. However, these approaches still require labeled data for training, and the interface for selecting the target class becomes cumbersome when we need a large number of classes to handle open-domain data. Wisdom et al. (2020) later proposed an unsupervised method called mixture invariant training (MixIT) for learning sound separation on noisy data. MixIT is designed to separate all sources at a time and also requires a post-selection process such as using a pre-trained sound classifier (Scott et al., 2021), which requires labeled data for training, to identify the target sounds. We summarize and compare related work in Table 1.

**Query-based sound separation**   Visual information has been used for selecting the target sound in speech (Ephrat et al., 2019; Afouras et al., 2020), music (Zhao et al., 2018; 2019; Tian et al., 2021) and universal sounds (Owens & Efros, 2018; Gao et al., 2018; Rouditchenko et al., 2019). While many image-queried sound separation approaches require clean video data that contains isolated sources, Tzinis et al. (2021) introduced an unsupervised method called AudioScope for separating on-screen sounds using noisy videos based on the MixIT model. While image queries can serve as a

| Method | Query type | Unlabeled data | Noisy data |
|---|:---:|:---:|:---:|
| USS (Kavalerov et al., 2019) | × | ✓ | |
| MixIT (Wisdom et al., 2020) | × | ✓ | ✓ |
| Universal Sound Selector (Ochiai et al., 2020) | Label | | |
| USS-Label (Kong et al., 2020) | Label | | (✓) |
| PixelPlayer (Zhao et al., 2018) | Image | ✓ | (✓) |
| AudioScope (Tzinis et al., 2021) | Image | ✓ | ✓ |
| SoundFilter (Gfeller et al., 2021) | Audio | ✓ | |
| Zero-shot audio separation (Chen et al., 2022) | Audio | | (✓) |
| Text-Queried (Liu et al., 2022) | Text | | |
| Text/Audio-Queried (Kilgour et al., 2022) | Text / Audio | | |
| CLIPSep (ours) | Text / Image | ✓ | |
| CLIPSep-NIT (ours) | Text / Image | ✓ | ✓ |

Table 1: Comparisons of related work in sound separation. '(✓)' indicates that the problem of noisy training data is partially addressed. CLIPSep-NIT denotes the proposed CLIPSep model trained with the noise invariant training. To the best of our knowledge, no previous work has attempted the problem of label-free text-queried sound separation.

natural interface for specifying the target sound in certain use cases, images of target sounds become unavailable in low-light conditions and for sounds from out-of-screen objects.

Another line of research uses the audio modality to query acoustically similar sounds. Chen et al. (2022) showed that such approach can generalize to unseen sounds. Later, Gfeller et al. (2021) cropped two disjoint segments from single recording and used them as a query-target pair to train a sound separation model, assuming both segments contain the same sound source. However, in many cases, it is impractical to prepare a reference audio sample for the desired sound as the query.

Most recently, text-queried sound separation has been studied as it provides a natural and scalable interface for specifying arbitrary target sounds as compared to systems that use a fixed set of class labels. Liu et al. (2022) employed a pretrained language model to encode the text query, and condition the model to separate the corresponding sounds. Kilgour et al. (2022) proposed a model that accepts audio or text queries in a hybrid manner. These approaches, however, require labeled text-audio paired data for training. Different from prior work, our goal is to learn text-queried sound separation for arbitrary sound *without* labeled data, specifically using unlabeled noisy videos in the wild.

**Contrastive language-image-audio pretraining** The CLIP model (Radford et al., 2021) has been used as a pretraining of joint embedding spaces among text, image and audio modalities for downstream tasks such as audio classification (Wu et al., 2022; Guzhov et al., 2022) and sound guided image manipulation (Lee et al., 2022). Pretraining is done either in a supervised manner using labels (Guzhov et al., 2022; Lee et al., 2022) or in a self-supervised manner by training an additional audio encoder to map input audio to the pretrained CLIP embedding space (Wu et al., 2022). In contrast, we explore the zero-shot modality transfer capability of the CLIP model by freezing the pre-trained CLIP model and directly optimizing the rest of the model for the target sound separation task.

## 3 METHOD

### 3.1 CLIPSEP—LEARNING TEXT-QUERIED SOUND SEPARATION WITHOUT LABELED DATA

In this section, we propose the CLIPSep model for text-queried sound separation without using labeled data. We base the CLIPSep model on Sound-of-Pixels (SOP) (Zhao et al., 2018) and replace the video analysis network of the SOP model. As illustrated in Figure 2, during training, the model takes as inputs an audio mixture $x = \sum_{i=1}^{n} s_i$, where $s_1, \ldots, s_n$ are the $n$ audio tracks, along with their corresponding images $y_1, \ldots, y_n$ extracted from the videos. We first transform the audio mixture $x$ into a magnitude spectrogram $X$ and pass the spectrogram through an audio U-Net (Ronneberger et al., 2015; Jansson et al., 2017) to produce $k (\geq n)$ intermediate masks $\tilde{M}_1, \ldots, \tilde{M}_k$. On the other stream, each image is encoded by the pretrained CLIP model (Radford et al., 2021) into an embedding $e_i \in \mathbb{R}^{512}$. The CLIP embedding $e_i$ will further be projected to a *query vector*

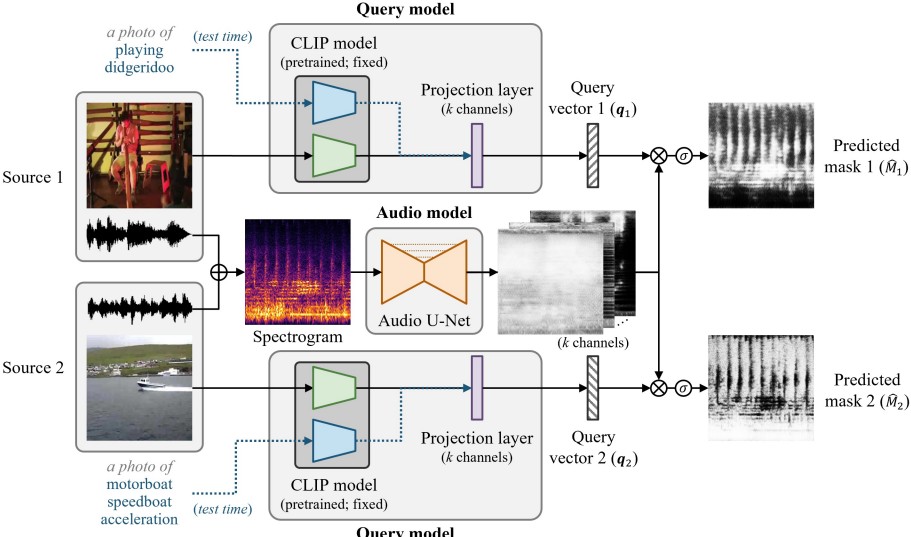

Figure 2: An illustration of the proposed CLIPSep model for $n = 2$. During training, we mix audio from two videos and train the model to separate each audio source given the corresponding video frame as the query. At test time, we instead use a text query in the form of "*a photo of [user input query]*" to query the sound separation model. Thanks to the properties of the pretrained CLIP model, the query vectors we obtain for the image and text queries are expected to be close.

$q_i \in \mathbb{R}^k$ by a *projection layer*, which is expected to extract only audio-relevant information from $e_i$.[2] Finally, the query vector $q_i$ will be used to mix the intermediate masks into the final predicted masks $\hat{M}_i = \sum_{j=1}^{k} \sigma(w_{ij} q_{ij} \tilde{M}_j + b_i)$, where $w_i \in \mathbb{R}^k$ is a learnable scale vector, $b_i \in \mathbb{R}$ a learnable bias, and $\sigma(\cdot)$ the sigmoid function. Now, suppose $M_i$ is the ground truth mask for source $s_i$. The training objective of the model is the sum of the weighted binary cross entropy losses for each source:

$$\mathcal{L}_{CLIPSep} = \sum_{i=1}^{n} WBCE(M_i, \hat{M}_i) = \sum_{i=1}^{n} X \odot \left( -M_i \log \hat{M}_i - (1 - M_i) \log \left( 1 - \hat{M}_i \right) \right). \quad (1)$$

At test time, thanks to the joint image-text embedding offered by the CLIP model, we feed a text query instead of an image to the query model to obtain the query vector and separate the target sounds accordingly (see Appendix A for an illustration). As suggested by Radford et al. (2021), we prefix the text query into the form of "*a photo of [user input query]*" to reduce the generalization gap.[3]

## 3.2 Noise invariant training—Handling noisy data in the wild

While the CLIPSep model can separate sounds given image or text queries, it assumes that the sources are clean and contain few query-irrelevant sounds. However, this assumption does not hold for videos in the wild as many of them contain out-of-screen sounds and various background noises. Inspired by the mixture invariant training (MixIT) proposed by Wisdom et al. (2020), we further propose the *noise invariant training* (NIT) to tackle the challenge of training with noisy data. As illustrated in Figure 3, we introduce $n$ additional permutation invariant heads called *noise heads* to the CLIPSep model, where the masks predicted by these heads are interchangeable during loss computation. Specifically, we introduce $n$ additional projection layers, and each of them takes as input the sum of all query vectors produced by the *query heads* (i.e., $\sum_{i=1}^{n} q_i$) and produce a vector that is later used to mix the intermediate masks into the predicted *noise mask*. In principle, the *query masks* produced by the query vectors are expected to extract query-relevant sounds due to their stronger correlations to their corresponding queries, while the interchangeable noise masks should 'soak up' other sounds.

---

[2]We extract three frames with 1-sec intervals and compute their mean CLIP embedding as the input to the projection layer to reduce the negative effects when the selected frame does not contain the objects of interest.

[3]Similar to how we prepare the image queries, we create four queries from the input text query using four query templates (see Appendix B) and take their mean CLIP embedding as the input to the projection layer.

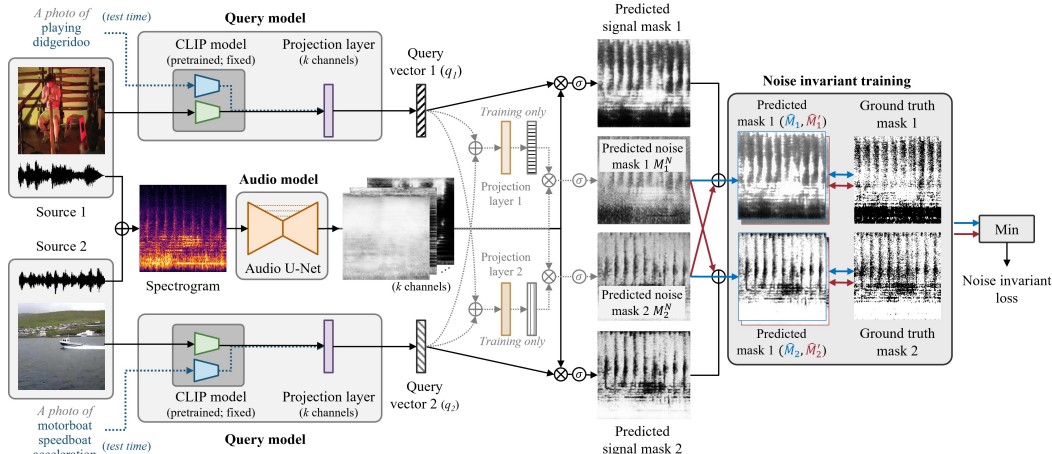

Figure 3: An illustration of the proposed CLIPSep-NIT model for $n = 2$. Similar to CLIPSep, we train the model to separate each audio source given the corresponding query image during training and switch to using a text query at test time. The two predicted noise masks are interchangeable for loss computation during training, and they are discarded at test time (grayed out paths).

Mathematically, let $M_1^Q, \ldots, M_n^Q$ be the predicted query masks and $M_1^N, \ldots, M_n^N$ be the predicted noise masks. Then, the noise invariant loss is defined as:

$$\mathcal{L}_{NIT} = \min_{(j_1,\ldots,j_n)\in\Sigma_n} \sum_{i=1}^n WBCE\left(M_i, \min\left(1, \hat{M}_i^Q + \hat{M}_{j_i}^N\right)\right), \tag{2}$$

where $\Sigma_n$ denotes the set of all permutations of $\{1, \ldots, n\}$.[4] Take $n = 2$ for example.[5] We consider the two possible ways for combining the query heads and the noise heads:

$$\text{(Arrangement 1)} \qquad \hat{M}_1 = \min\left(1, \hat{M}_1^Q + \hat{M}_1^N\right), \quad \hat{M}_2 = \min\left(1, \hat{M}_2^Q + \hat{M}_2^N\right), \tag{3}$$

$$\text{(Arrangement 2)} \qquad \hat{M}_1' = \min\left(1, \hat{M}_1^Q + \hat{M}_2^N\right), \quad \hat{M}_2' = \min\left(1, \hat{M}_2^Q + \hat{M}_1^N\right). \tag{4}$$

Then, the noise invariant loss is defined as the smallest loss achievable:

$$\mathcal{L}_{NIT}^{(2)} = \min\left(WBCE\left(M_1, \hat{M}_1\right) + WBCE\left(M_2, \hat{M}_2\right), WBCE\left(M_1, \hat{M}_1'\right) + WBCE\left(M_2, \hat{M}_2'\right)\right). \tag{5}$$

Once the model is trained, we discard the noise heads and use only the query heads for inference (see Appendix A for an illustration). Unlike the MixIT model (Wisdom et al., 2020), our proposed noise invariant training still allows us to specify the target sound by an input query, and it does not require any post-selection process as we only use the query heads during inference.

In practice, we find that the model tends to assign part of the target sounds to the noise heads as these heads can freely enjoy the optimal permutation to minimize the loss. Hence, we further introduce a regularization term to penalize producing high activations on the noise masks:

$$\mathcal{L}_{REG} = \max\left(0, \sum_{i=1}^n \text{mean}\left(\hat{M}_i^N\right) - \gamma\right), \tag{6}$$

where $\gamma \in [0, n]$ is a hyperparameter that we will refer to as the *noise regularization level*. The proposed regularization has no effect when the sum of the means of all the noise masks is lower than a predefined threshold $\gamma$, while having a linearly growing penalty when the sum is higher than $\gamma$. Finally, the training objective of the CLIPSep-NIT model is a weighted sum of the noise invariant loss and regularization term: $\mathcal{L}_{CLIPSep-NIT} = \mathcal{L}_{NIT} + \lambda \mathcal{L}_{REG}$, where $\lambda \in \mathbb{R}$ is a weight hyperparameter. We set $\lambda = 0.1$ for all experiments, which we find work well across different settings.

---

[4]We note that CLIPSep-NIT considers $2n$ sources in total as the model has $n$ queried heads and $n$ noise heads. While PIT (Yu et al., 2017) and MixIT (Wisdom et al., 2020) respectively require $\mathcal{O}((2n)!)$ and $\mathcal{O}(2^{2n})$ search to consider $2n$ sources, the proposed NIT only requires $\mathcal{O}(n!)$ permutation in the loss computation.

[5]Since our goal is not to further separate the noise into individual sources but to separate the sounds that correspond to the query, $n$ may not need to be large. In practice, we find that the CLIPSep-NIT model with $n = 2$ already learns to handle the noise properly and can successfully transfer to the text-queried mode. Thus, we use $n = 2$ throughout this paper and leave the testing on larger $n$ as future work.

| Model | Unlabeled data | Post-proc. free | Query type | | SDR [dB] | |
| --- | --- | --- | --- | --- | --- | --- |
| | | | Training | Test | Mean | Median |
| Mixture | - | - | - | - | $0.00 \pm 0.89$ | 0.00 |
| **Text-queried models** | | | | | | |
| CLIPSep | ✓ | ✓ | Image | Text | **$5.49 \pm 0.72$** | **4.97** |
| CLIPSep-Text | | ✓ | Text | Text | $7.91 \pm 0.81$ | 7.46 |
| CLIPSep-Hybrid | | ✓ | Text + Image | Text | **$8.36 \pm 0.83$** | **8.72** |
| **Image-queried models** | | | | | | |
| SOP (Zhao et al., 2018) | ✓ | ✓ | Image | Image | $6.59 \pm 0.85$ | **6.22** |
| CLIPSep | ✓ | ✓ | Image | Image | **$7.03 \pm 0.70$** | 5.85 |
| CLIPSep-Text | | ✓ | Text | Image | $6.25 \pm 0.72$ | 6.19 |
| CLIPSep-Hybrid | | ✓ | Text + Image | Image | **$8.06 \pm 0.79$** | **8.01** |
| **Nonqueried models** | | | | | | |
| LabelSep | | ✓ | Label | Label | $8.18 \pm 0.80$ | **7.82** |
| PIT (Yu et al., 2017) | ✓ | | × | × | **$8.68 \pm 0.76$** | 7.67 |

Table 2: Results on the MUSIC dataset. Standard errors are reported in the mean SDR column. Bold values indicate the largest SDR achieved per group.

## 4 EXPERIMENTS

We base our implementations on the code provided by Zhao et al. (2018) (`https://github.com/hangzhaomit/Sound-of-Pixels`). Implementation details can be found in Appendix C.

### 4.1 EXPERIMENTS ON CLEAN DATA

We first evaluate the proposed CLIPSep model without the noise invariant training on musical instrument sound separation task using the MUSIC dataset, as done in (Zhao et al., 2018). This experiment is designed to focus on evaluating the quality of the learned query vectors and the zero-shot modality transferability of the CLIPSep model on a small, clean dataset rather than showing its ability to separate arbitrary sounds. The MUSIC dataset is a collection of 536 video recordings of people playing a musical instrument out of 11 instrument classes. Since no existing work has trained a text-queried sound separation model using only unlabeled data to our knowledge, we compare the proposed CLIPSep model with two baselines that serve as upper bounds—the PIT model (Yu et al., 2017, see Appendix D for an illustration) and a version of the CLIPSep model where the query model is replaced by learnable embeddings for the labels, which we will refer to as the LabelSep model. In addition, we also include the SOP model (Zhao et al., 2018) to investigate the quality of the query vectors as the CLIPSep and SOP models share the same network architecture except the query model.

We report the results in Table 2. Our proposed CLIPSep model achieves a mean signal-to-distortion ratio (SDR) (Vincent et al., 2006) of 5.49 dB and a median SDR of 4.97 dB using text queries in a zero-shot modality transfer setting. When using image queries, the performance of the CLIPSep model is comparable to that of the SOP model. This indicates that the CLIP embeddings are as informative as those produced by the SOP model. The performance difference between the CLIPSep model using text and image queries at test time indicates the *zero-shot modality transfer gap*. We observe 1.54 dB and 0.88 dB differences on the mean and median SDRs, respectively. Moreover,

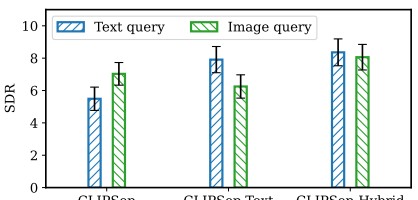

Figure 4: Mean SDR and standard errors of the models trained and tested on different modalities.

we also report in Table 2 and Figure 4 the performance of the CLIPSep models trained on different modalities to investigate their modality transferability in different settings. We notice that when we train the CLIPSep model using text queries, dubbed as CLIPSep-Text, the mean SDR using text queries increases to 7.91 dB. However, when we test this model using image queries, we observe a 1.66 dB difference on the mean SDR as compared to that using text queries, which is close to

| Model | Unlabeled data | Post-proc. free | MUSIC$^+$ | | VGGSound-Clean$^+$ | |
|---|---|---|---|---|---|---|
| | | | Mean SDR | Median SDR | Mean SDR | Median SDR |
| Mixture | - | - | $4.49 \pm 1.41$ | 2.04 | $-0.77 \pm 1.31$ | -0.84 |
| **Text-queried models** | | | | | | |
| CLIPSep | ✓ | ✓ | $9.71 \pm 1.21$ | 8.73 | $2.76 \pm 1.00$ | **3.95** |
| CLIPSep-NIT | ✓ | ✓ | **$10.27 \pm 1.04$** | **10.02** | **$3.05 \pm 0.73$** | 3.26 |
| BERTSep | | ✓ | $4.67 \pm 0.44$ | 4.41 | $5.09 \pm 0.80$ | 5.49 |
| CLIPSep-Text | | ✓ | $10.73 \pm 0.99$ | 9.93 | $5.49 \pm 0.82$ | 5.06 |
| **Image-queried models** | | | | | | |
| SOP (Zhao et al., 2018) | ✓ | ✓ | $11.44 \pm 1.18$ | 11.18 | $2.99 \pm 0.84$ | 3.89 |
| CLIPSep | ✓ | ✓ | **$12.20 \pm 1.17$** | **12.42** | **$5.46 \pm 0.79$** | **5.35** |
| CLIPSep-NIT | ✓ | ✓ | $11.28 \pm 1.08$ | 10.83 | $4.84 \pm 0.66$ | 3.57 |
| CLIPSep-Text | | ✓ | $9.89 \pm 1.04$ | 8.09 | $2.45 \pm 0.70$ | 1.74 |
| **Nonqueried models** | | | | | | |
| PIT (Yu et al., 2017) | ✓ | | **$12.24 \pm 1.20$** | **12.53** | **$5.73 \pm 0.79$** | **4.97** |
| LabelSep | | ✓ | - | - | $5.55 \pm 0.81$ | 5.29 |

Table 3: Results of the MUSIC$^+$ and VGGSound-Clean$^+$ evaluations (see Section 4.2). Standard errors are reported in the mean SDR [dB] columns. Bold values indicate the largest SDR achieved per group. We use $\gamma = 0.25$ for CLIPSep-NIT. Note that the LabelSep model does not work on the MUSIC dataset due to the different label taxonomies of the MUSIC and VGGSound datasets.

the mean SDR difference we observe for the model trained with image queries. Finally, we train a CLIPSep model using both text and image queries in alternation, dubbed as CLIPSep-Hybrid. We see that it leads to the best test performance for both text and image modalities, and there is only a mean SDR difference of 0.30 dB between using text and image queries. As a reference, the LabelSep model trained with labeled data performs worse than the CLIPSep-Hybrid model using text queries. Further, the PIT model achieves a mean SDR of 8.68 dB and a median SDR of 7.67 dB, but it requires post-processing to figure out the correct assignments.

## 4.2 EXPERIMENTS ON NOISY DATA

Next, we evaluate the proposed method on a large-scale dataset aiming at universal sound separation. We use the VGGSound dataset (Chen et al., 2020), a large-scale audio-visual dataset containing more than 190,000 10-second videos in the wild out of more than 300 classes. We find that the audio in the VGGSound dataset is often noisy and contains off-screen sounds and background noise. Although we train the models on such noisy data, it is not suitable to use the noisy data as targets for evaluation because it fails to provide reliable results. For example, if the target sound labeled as "dog barking" also contains human speech, separating only the dog barking sound provides a lower SDR value than separating the mixture of dog barking sound and human speech even though the text query is "dog barking". (Note that we use the labels only for evaluation but not for training.) To avoid this issue, we consider the following two evaluation settings:

- **MUSIC$^+$**: Samples in the MUSIC dataset are used as clean targets and mixed with a sample in the VGGSound dataset as an interference. The separation quality is evaluated on the clean target from the MUSIC dataset. As we do not use the MUSIC dataset for training, this can be considered as zero-shot transfer to a new data domain containing unseen sounds (Radford et al., 2019; Brown et al., 2020). To avoid the unexpected overlap of the target sound types in the MUSIC and VGGSound datasets caused by the label mismatch, we exclude all the musical instrument playing videos from the VGGSound dataset in this setting.

- **VGGSound-Clean$^+$**: We manually collect 100 clean samples that contain distinct target sounds from the VGGSound test set, which we will refer to as VGGSound-Clean. We mix an audio sample in VGGSound-Clean with another in the test set of VGGSound. Similarly, we consider the VGGSound audio as an interference sound added to the relatively cleaner VGGSound-Clean audio and evaluate the separation quality on the VGGSound-Clean stem.

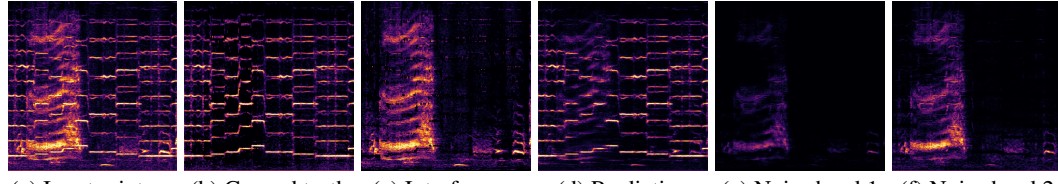

|(a) Input mixture|(b) Ground truth|(c) Interference|(d) Prediction|(e) Noise head 1|(f) Noise head 2|

Figure 5: Example results of the proposed CLIPSep-NIT model with $\gamma = 0.25$ on the MUSIC$^+$ dataset. We mix the an audio sample ("violin" in this example) in the MUSIC dataset with an interference audio sample ("people sobbing" in this example) in the VGGSound dataset to create an artificial mixture. (b) and (c) show the reconstructed signals using the ground truth ideal binary masks. The spectrograms are shown in the log frequency scale. We observe that the proposed model successfully separates the desired sounds (i.e., (d)) from query-irrelevant noise (i.e., (e) and (f)).

Table 3 shows the evaluation results. First, CLIPSep successfully learns text-queried sound separation even with noisy unlabeled data, achieving 5.22 dB and 3.53 dB SDR improvements over the mixture on MUSIC$^+$ and VGGSound-Clean$^+$, respectively. By comparing CLIPSep and CLIPSep-NIT, we observe that NIT improves the mean SDRs in both settings. Moreover, on MUSIC$^+$, CLIPSep-NIT's performance matches that of CLIPSep-Text, which utilizes labels for training, achieving only a 0.46 dB lower mean SDR and even a 0.05 dB higher median SDR. This result suggests that the proposed self-supervised text-queried sound separation method can learn separation capability competitive with the fully supervised model in some target sounds. In contrast, there is still a gap between them on VGGSound-Clean$^+$, possibly because the videos of non-music-instrument objects are more noisy in both audio and visual domains, thus resulting in a more challenging zero-shot modality transfer. This hypothesis is also supported by the higher zero-shot modality transfer gap (mean SDR difference of image- and text-queried mode) of 1.79 dB on VGGSound-Clean$^+$ than that of 1.01 dB on MUSIC$^+$ for CLIPSep-NIT. In addition, we consider another baseline model that replaces the CLIP model in CLIPSep with a BERT encoder (Devlin et al., 2019), which we call BERTSep. Interestingly, although BERTSep performs similarly to CLIPSep-Text on VGGSound-Clean$^+$, the performance of BERTSep is significantly lower than that of CLIPSep-Text on MUISC$^+$, indicating that BERTSep fails to generalize to unseen text queries. We hypothesize that the CLIP text embedding captures the timbral similarity of musical instruments better than the BERT embedding do, because the CLIP model is aware of the visual similarity between musical instruments during training. Moreover, it is interesting to see that CLIPSep outperforms CLIPSep-NIT when an image query is used at test time (domain-matched condition), possibly because images contain richer context information such as objects nearby and backgrounds than labels, and the models can use such information to better separate the target sound. While CLIPSep has to fully utilize such information, CLIPSep-NIT can use the noise heads to model sounds that are less relevant to the image query. Since we remove the noise heads from CLIPSep-NIT during the evaluation, it can rely less on such information from the image, thus improving the zero-shot modality transferability. Figure 5 shows an example of the separation results on MUSIC$^+$ (see Figures 12 to 15 for more examples). We observe that the two noise heads contain mostly background noise. Audio samples can be found on our demo website.[1]

## 4.3 EXAMINING THE EFFECTS OF THE NOISE REGULARIZATION LEVEL $\gamma$

In this experiment, we examine the effects of the noise regularization level $\gamma$ in Equation (6) by changing the value from 0 to 1. As we can see from Figure 6 (a) and (b), CLIPSep-NIT with $\gamma = 0.25$ achieves the highest SDR on both evaluation settings. This suggests that the optimal $\gamma$ value is not sensitive to the evaluation dataset. Further, we also report in Figure 6 (c) the total mean noise head activation, $\sum_{i=1}^{n} \text{mean}(\hat{M}_i^N)$, on the validation set. As $\hat{M}_i^N$ is the mask estimate for the noise, the total mean noise head activation value indicates to what extent signals are assigned to the noise head. We observe that the proposed regularizer successfully keeps the total mean noise head activation close to the desired level, $\gamma$, for $\gamma \leq 0.5$. Interestingly, the total mean noise head activation is still around 0.5 when $\gamma = 1.0$, suggesting that the model inherently tries to use both the query-heads and the noise heads to predict the noisy target sounds. Moreover, while we discard the noise heads during evaluation in our experiments, keeping the noise heads can lead to a higher SDR as shown in

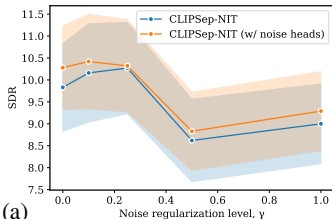 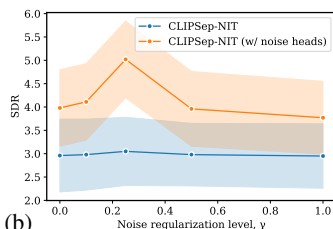 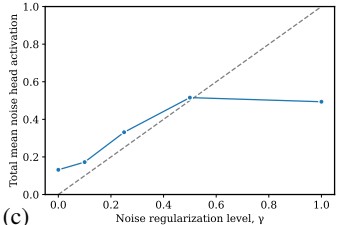

(a)  (b)  (c)

Figure 6: Effects of the noise regularization level $\gamma$ for the proposed CLIPSep-NIT model—mean SDR for the (a) MUSIC$^+$ and (b) VGGSound-Clean$^+$ evaluations, and (c) the total mean noise head activation, $\sum_{i=1}^{n} \mathrm{mean}(\hat{M}_i^N)$, on the validation set. The shaded areas show standard errors.

Figure 6 (a) and (b), which can be helpful in certain use cases where a post-processing procedure similar to the PIT model (Yu et al., 2017) is acceptable.

## 5 DISCUSSIONS

For the experiments presented in this paper, we work on labeled datasets so that we can evaluate the performance of the proposed models. However, our proposed models do not require any labeled data for training, and can thus be trained on larger unlabeled video collections in the wild. Moreover, we observe that the proposed model shows the capability of combing multiple queries, e.g., "*a photo of [query A] and [query B]*," to extract multiple target sounds, and we report the results on the demo website. This offers a more natural user interface against having to separate each target sound and mix them via an additional post-processing step. We also show in Appendix G that our proposed model is robust to different text queries and can extract the desired sounds.

In our experiments, we often observe a modality transfer gap greater than 1 dB difference of SDR. A future research direction is to explore different approaches to reduce the modality transfer gap. For example, the CLIP model is pretrained on a different dataset, and thus finetuning the CLIP model on the target dataset can help improve the underlying modality transferability within the CLIP model. Further, while the proposed noise invariant training is shown to improve the training on noisy data and reduce the modality transfer gap, it still requires a sufficient audio-visual correspondence for training video. In other words, if the audio and images are irrelevant in most videos, the model will struggle to learn the correspondence between the query and target sound. In practice, we find that the data in the VGGSound dataset often contains off-screen sounds and the labels sometimes correspond to only part of the video content. Hence, filtering on the training data to enhance its audio-visual correspondence can also help reduce the modality transfer gap. This can be achieved by self-supervised audio-visual correspondence prediction (Arandjelović & Zisserman, 2017a;b) or temporal synchronization (Korbar et al., 2018; Owens & Efros, 2018).

Another future direction is to explore the semi-supervised setting where a small subset of labeled data can be used to improve the modality transferability. We can also consider the proposed method as a pretraining on unlabeled data for other separation tasks in the low-resource regime. We include in Appendix H a preliminary experiment in this aspect using the ESC-50 dataset (Piczak, 2015).

## 6 CONCLUSION

In this work, we have presented a novel text-queried universal sound separation model that can be trained on noisy unlabeled videos. In this end, we have proposed to use the contrastive image-language pretraining to bridge the audio and text modalities, and proposed the noise invariant training for training a query-based sound separation model on noisy data. We have shown that the proposed models can learn to separate an arbitrary sound specified by a text query out of a mixture, even achieving competitive performance against a fully supervised model in some settings. We believe our proposed approach closes the gap between the ways humans and machines learn to focus on a sound in a mixture, namely, the multi-modal self-supervised learning paradigm of humans against the supervised learning paradigm adopted by existing label-based machine learning approaches.

ACKNOWLEDGEMENTS

We would like to thank Stefan Uhlich, Giorgio Fabbro and Woosung Choi for their helpful comments during the preparation of this manuscript. We also thank Mayank Kumar Singh for supporting the setup of the subjective test in Appendix F. Hao-Wen thank J. Yang and Family Foundation and Taiwan Ministry of Education for supporting his PhD study.

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

## A    INFERENCE PIPELINE OF CLIPSEP AND CLIPSEP-NIT

Figure 7 illustrates the inference pipeline for the proposed CLIPSep and CLIPSep-NIT models. For the CLIPSep-NIT model, we discard the noise heads at test time. The masked spectrogram is combined with the input phase and converted to the waveform by the inverse short-time Fourier transform (STFT).

## B    QUERY ENSEMBLING

Radford et al. (2021) suggest that using a *prompt template* in the form of "*a photo of [user input query]*" helps bridge the distribution gap between text queries used for zero-shot image classification and text in the training dataset for the CLIP model. They further show that the ensemble of various prompt templates improve the generalizability. Motivated by this observation, we adopt a similar idea and use several query templates at test time (see Table 4). These query templates are heuristically chosen to handle the noisy images extracted from videos.

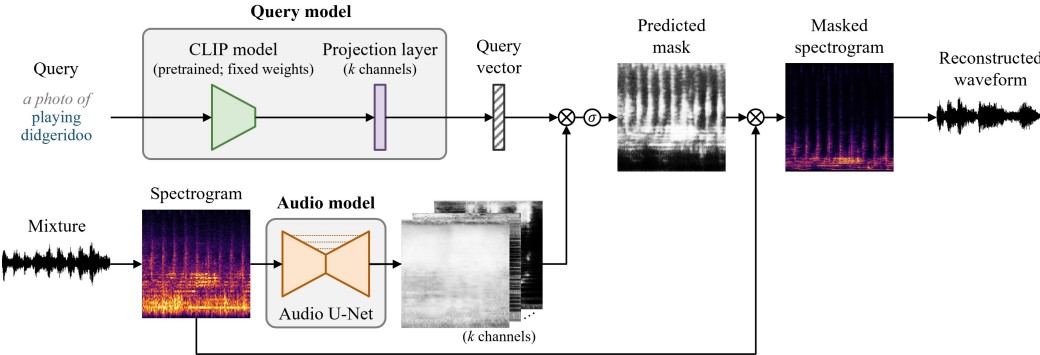

Figure 7: Inference pipeline of the proposed CLIPSep and CLIPSep-NIT models. Note that the mixture spectrogram and the masks are shown in log frequency scale, while the masked spectrogram is shown in linear frequency scale.

| Query template | Example query for the label "*dog barking*" |
|---|---|
| a photo of [*user input query*] | a photo of dog barking |
| a photo of the small [*user input query*] | a photo of the small dog barking |
| a low resolution photo of a [*user input query*] | a low resolution photo of a dog barking |
| a photo of many [*user input query*] | a photo of many dog barking |

Table 4: Query templates used in our experiments.

## C  IMPLEMENTATION DETAILS

We implement the audio model as a 7-layer U-Net (Ronneberger et al., 2015). We use $k = 32$. We use binary masks as the ground truth masks during training while using the raw, real-valued masks for evaluation. We train all the models for 200,000 steps with a batch size of 32. We use the Adam optimizer (Kingma & Ba, 2015) with $\beta_1 = 0.9$, $\beta_2 = 0.999$ and $\epsilon = 10^{-8}$. In addition, we clip the norm of the gradients to 1.0 (Zhang et al., 2020). We adopt the following learning rate schedule with a warm-up—the learning rate starts from 0 and grows to 0.001 after 5,000 steps, and then it linearly drops to 0.0001 at 100,000 steps and keeps this value thereafter. We validate the model every 10,000 steps using image queries as we do not assume labeled data is available for the validation set. We use a sampling rate of 16,000 Hz and work on audio clips of length 65,535 samples ($\approx$ 4 seconds). During training, we randomly sample a center frame from a video and extract three frames (images) with 1-sec intervals and 4-sec audio around the center frame. During inference, for image-queried models, we extract three frames with 1-sec intervals around the center of the test clip. For the spectrogram computation, we use a filter length of 1024, a hop length of 256 and a window size of 1024 in the short-time Fourier transform (STFT). We resize images extracted from video to a size of 224-by-224 pixels. For the CLIPSep-Hybrid model, we alternatively train the model with text and image queries, i.e., one batch with all image queries and next with all text queries, and so on. We implement all the models using the PyTorch library (Paszke et al., 2019). We compute the signal-to-distortion ratio (SDR) using museval (Stöter et al., 2018).

In our preliminary experiments, we also tried directly predicting the final mask by conditioning the audio model on the query vector. We applied this modification for both SOP and CLIPSep models, however, we observe that this architecture is prone to overfitting. We hypothesize that this is because the audio model is powerful enough to remember the subtle clues in the query vector, which hinder the generalization to a new sound and query. In contrast, the proposed architecture first predicts over-determined masks and then combines them on the basis of the query vector, which avoids the overfitting problem due to the simple fusion step.

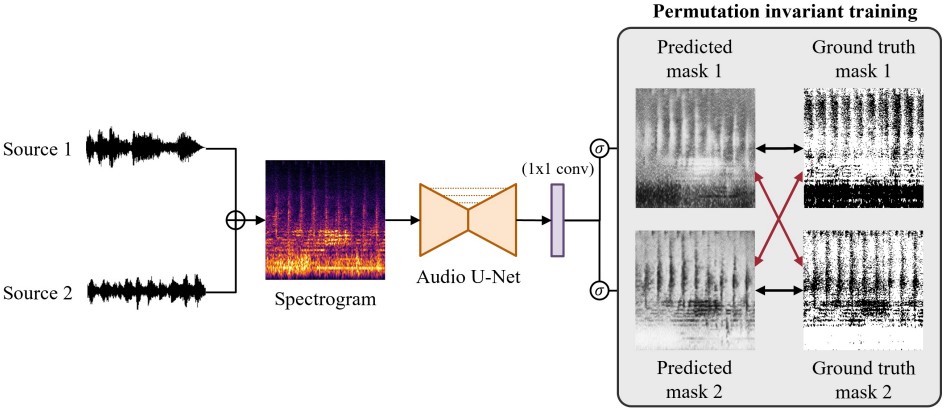

Figure 8: An illustration of the PIT model for $n = 2$. The two predicted masks are interchangeable during the loss computation. Since the two predicted masks are interchangeable, the PIT model requires an additional post-selection step to obtain the target sound.

## D    Permutation Invariant Training

Figure 8 illustrates the permuatation invariant training (PIT) model (Yu et al., 2017). The permutation invariant loss is defined as follows for $n = 2$.

$$\mathcal{L}_{PIT} = \min \left( WBCE(M_1, \hat{M}_1) + WBCE(M_2, \hat{M}_2), WBCE(M_1, \hat{M}_2) + WBCE(M_2, \hat{M}_1) \right), \quad (7)$$

where $\hat{M}_1$ and $\hat{M}_2$ are the predicted masks. Note that the PIT model requires an additional post-selection step to obtain the target sound.

## E    Qualitative Example Results

We show in Figures 12 to 15 some example results. More results and audio samples can be found at https://sony.github.io/CLIPSep/.

## F    Subjective Evaluation

We conduct a subjective test to evaluate whether the SDR results aligned with perceptual quality. As done in the Sound of Pixel (Zhao et al., 2018), separated audio samples are randomly presented to evaluators, and the following question is asked: "Which sound do you hear? 1. A, 2. B, 3. Both, or 4. None of them". Here A and B are replaced by labels of their mixture sources, e.g. A=accordion, B=engine accelerating. Ten samples (including naturally occurring mixture) are evaluated for each model and 16 evaluators have participated in the evaluation. Table 5 shows the percentages of samples which are correctly identified the target sound class (Correct), which are incorrectly identified the target sound sources (Wrong), which are selected as both sounds are audible (Both), and which are selected as neither of the sounds are audible (None). The results indicate that the evaluators more often choose the correct sound source for CLIPSep-NIT (83.8%) than CLIPSep (66.3%) with text queries. Notably, CLIPSep-NIT with text-query obtained a higher correct score than that with image-query, which matches the training mode. This is probably because image queries often contain information about backgrounds and environments, hence, some noise and off-screen sounds are also suggested by the image-queries and leak to the query head. In contrast, text-queries purely contain the information of target sounds, thus, the query head more aggressively extract the target sounds.

## G    Robustness to different queries

To examine the model's robustness to different queries, we take the same input mixture and query the model with different text queries. We use the CLIPSep-NIT model on the MUSIC$^+$ dataset and

| Model | Query type | Correct [%] | Wrong [%] | Both [%] | None [%] |
|-------|-----------|------------|-----------|----------|----------|
| Mixture | - | 17.5 | 10.0 | 72.5 | 0.0 |
| CLIPSep | Image | 70.6 | 0.0 | 29.4 | 0.0 |
|  | Text | 66.3 | 3.8 | 30.0 | 0.0 |
| CLIPSep-NIT | Image | 68.8 | 1.9 | 28.1 | 1.3 |
|  | Text | 83.8 | 0.6 | 15.0 | 0.6 |

Table 5: Subjective test results.

| Model | Post-processing free | Dataset | | SDR | |
|-------|----------------------|---------|---|-----|---|
|  |  | Training | Finetuning | Mean | Median |
| Mixture | - | - | - | $0.00 \pm 0.44$ | 0.00 |
| PIT | × | VGGSound | - | $4.90 \pm 0.26$ | 2.44 |
| CLIPSep | ✓ | VGGSound | - | $1.07 \pm 0.28$ | 2.34 |
|  | ✓ | ESC-50 | - | $5.18 \pm 0.26$ | 5.09 |
|  | ✓ | VGGSound | ESC-50 | $6.73 \pm 0.26$ | 5.89 |

Table 6: Results on the ESC-50 dataset. Standard errors are reported in the mean SDR column.

report in Figure 16 the results. We see that the model is robust to different text queries and can extract the desired sounds. Audio samples can be found at https://sony.github.io/CLIPSep/.

## H   FINETUNING EXPERIMENTS ON THE ESC-50 DATASET

In this experiment, we aim to examine the possibilities of having a clean dataset for further finetuning. We consider the ESC-50 dataset (Piczak, 2015), a collection of 2,000 high-quality environmental audio recordings, as the clean dataset here.[6] We report the experimental results in Table 6. We can see that the model pretrained on VGGSound does not generalize well to the ESC-50 dataset as the ESC-50 contains much cleaner sounds, i.e., without query-irrelevant sounds and background noise. Further, if we train the CLIPSep model from scratch on the ESC-50 dataset, it can only achieve a mean SDR of 5.18 dB and a median SDR of 5.09 dB. However, if we take the model pretrained on the VGGSound dataset and finetune it on the ESC-50 dataset, it can achieve a mean SDR of 6.73 dB and a median SDR of 4.89 dB, resulting in an improvement of 1.55 dB on the mean SDR.

## I   TRAINING BEHAVIORS

We present in Figure 9 the training and validation losses along the training progress. Please note that we only show the results obtained using text queries for reference but do not use them for choosing the best model. We also evaluate the intermediate checkpoints every 10,000 steps and present in Figure 10 the test SDR along the training progress. In addition, for the CLIPSep-NIT model, we visualize in Figure 11 the total mean noise head activation, $\sum_{i=1}^{n} \text{mean}(\hat{M}_i^N)$, along the training progress. We can see that the total mean noise head activation stays around the desired level for $\gamma = 0.1, 0.25$. For $\gamma = 0.5$ and the unregularized version, the total mean noise head activation converges to a similar value around 0.55.

---

[6] https://github.com/karolpiczak/ESC-50

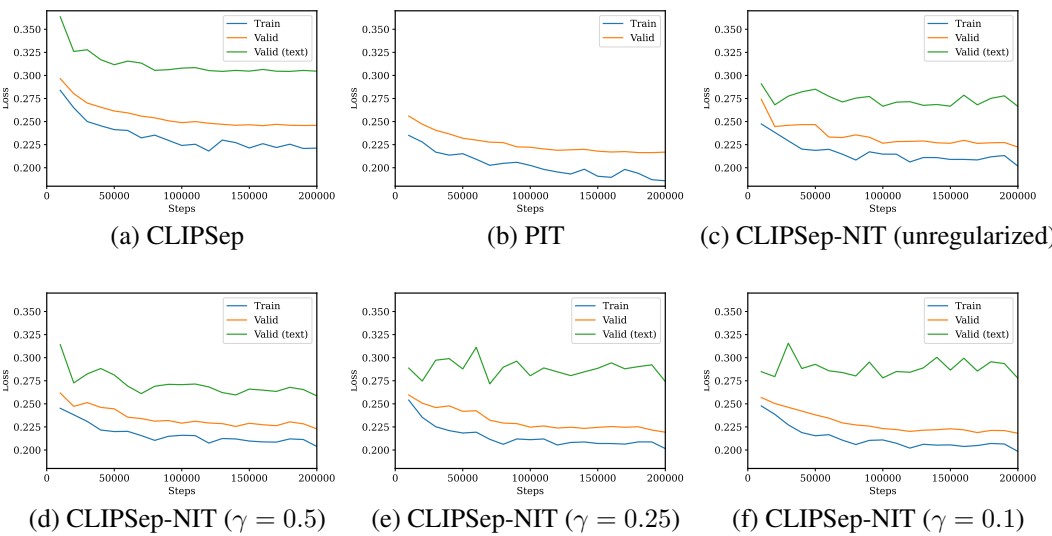

Figure 9: Training and validation losses along the training progress on the VGGSound dataset. We also include the losses computed using text queries instead of image queries. The y-axes are intentionally set to the same range for easy comparison. Note that we do not use the validation results obtained with text queries for choosing the best model.

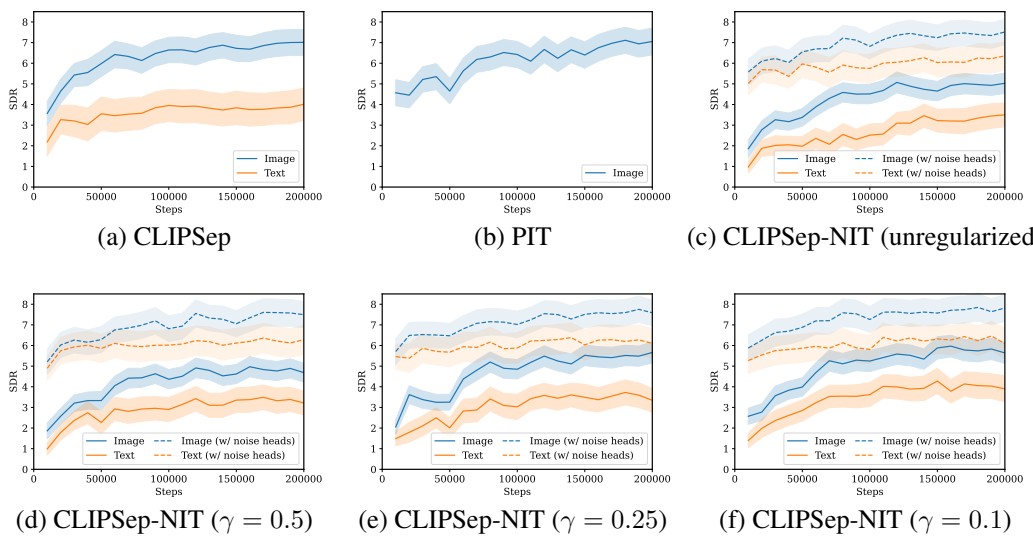

Figure 10: Test SDR along the training progress on the VGGSound-Clean dataset. The y-axes are intentionally set to the same range for easy comparison.

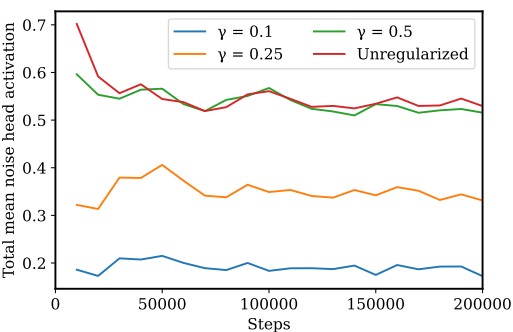

Figure 11: Total mean noise head activation, $\sum_{i=1}^{n} \operatorname{mean}(\hat{M}_i^N)$, on the validation set for the CLIPSep-NIT models along the training progress.

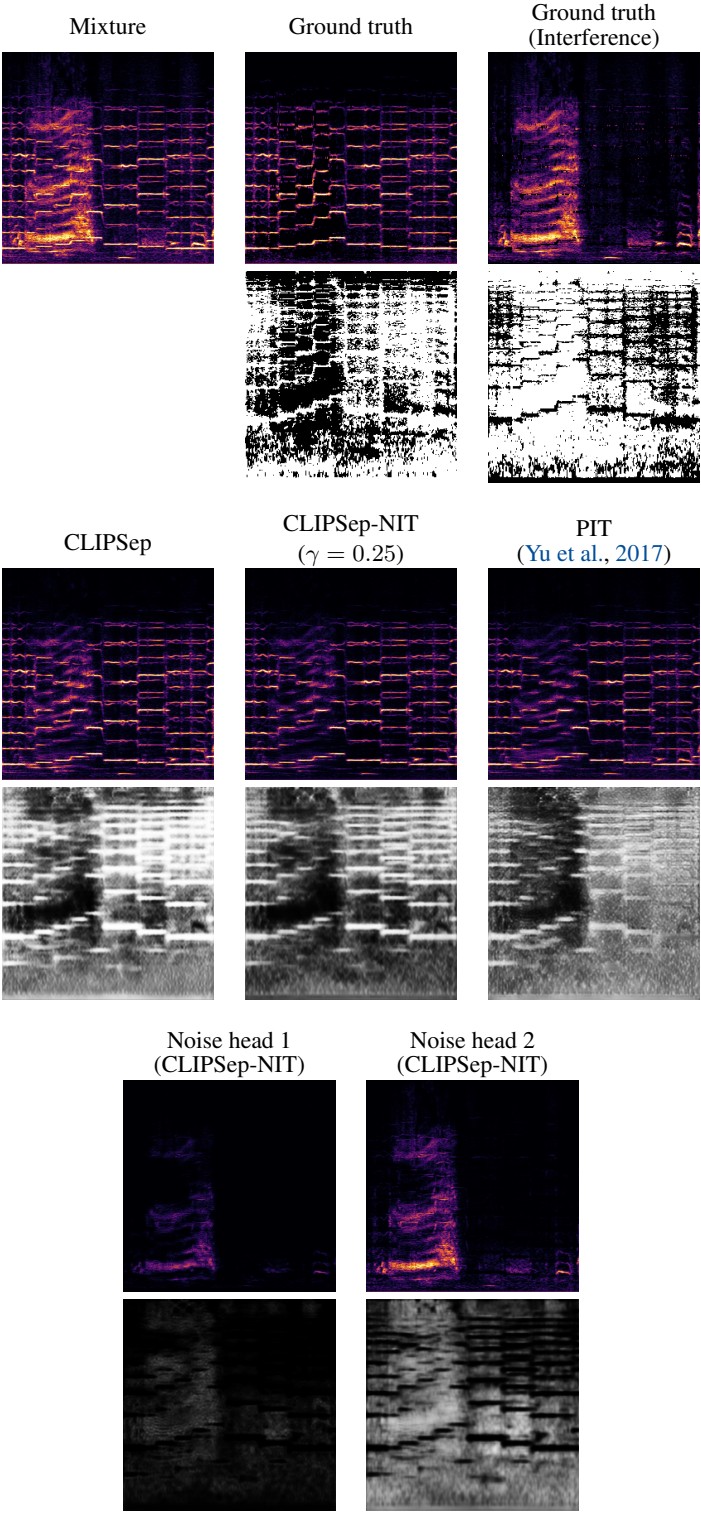

Figure 12: Example results on the MUSIC$^+$ dataset. Target source—"violin"; interference—"people sobbing"; query—"*violin*". The spectrograms and masks are shown in the log and linear frequency scales, respectively.

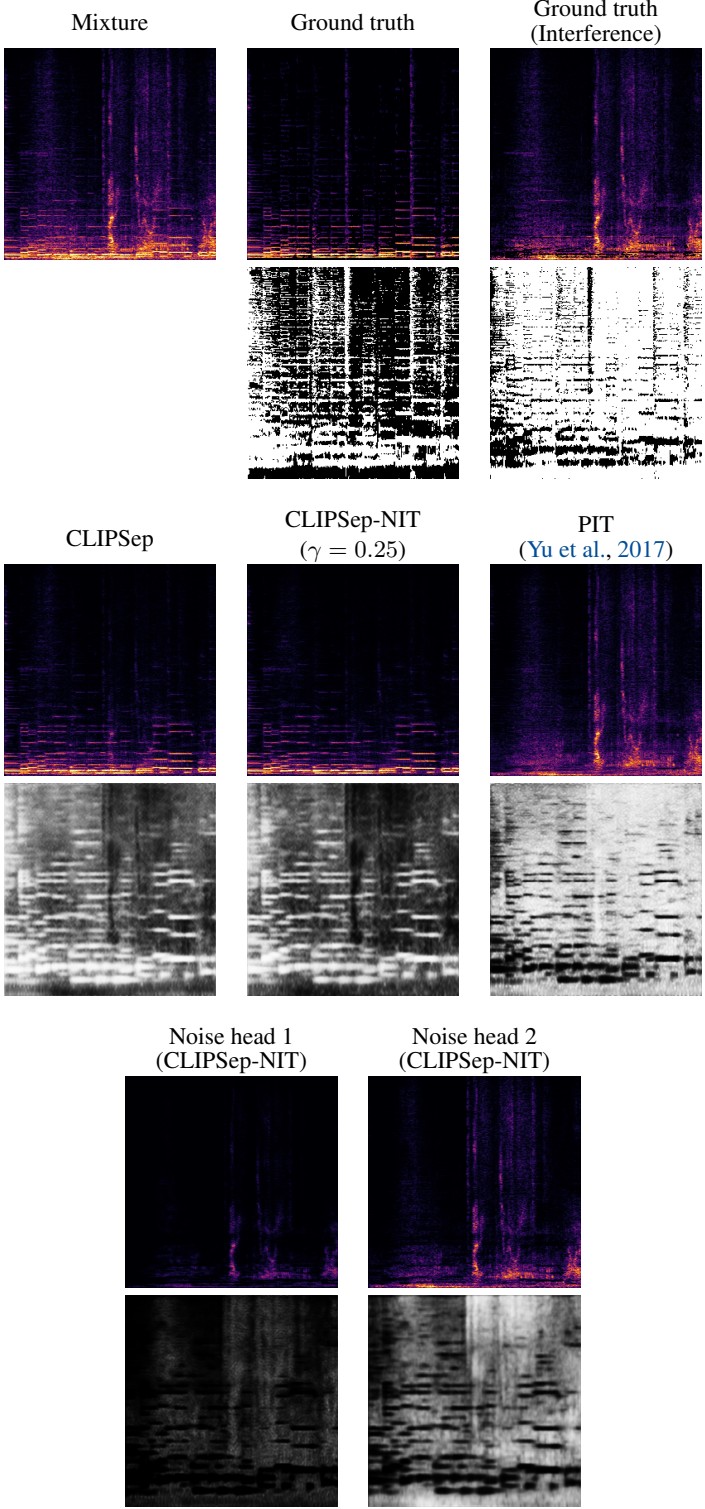

Figure 13: Example results on the MUSIC$^+$ dataset. Target source—"acoustic guitar"; interference—"cheetah chirrup", query—"*acoustic guitar*". The spectrograms and masks are shown in the log and linear frequency scales, respectively.

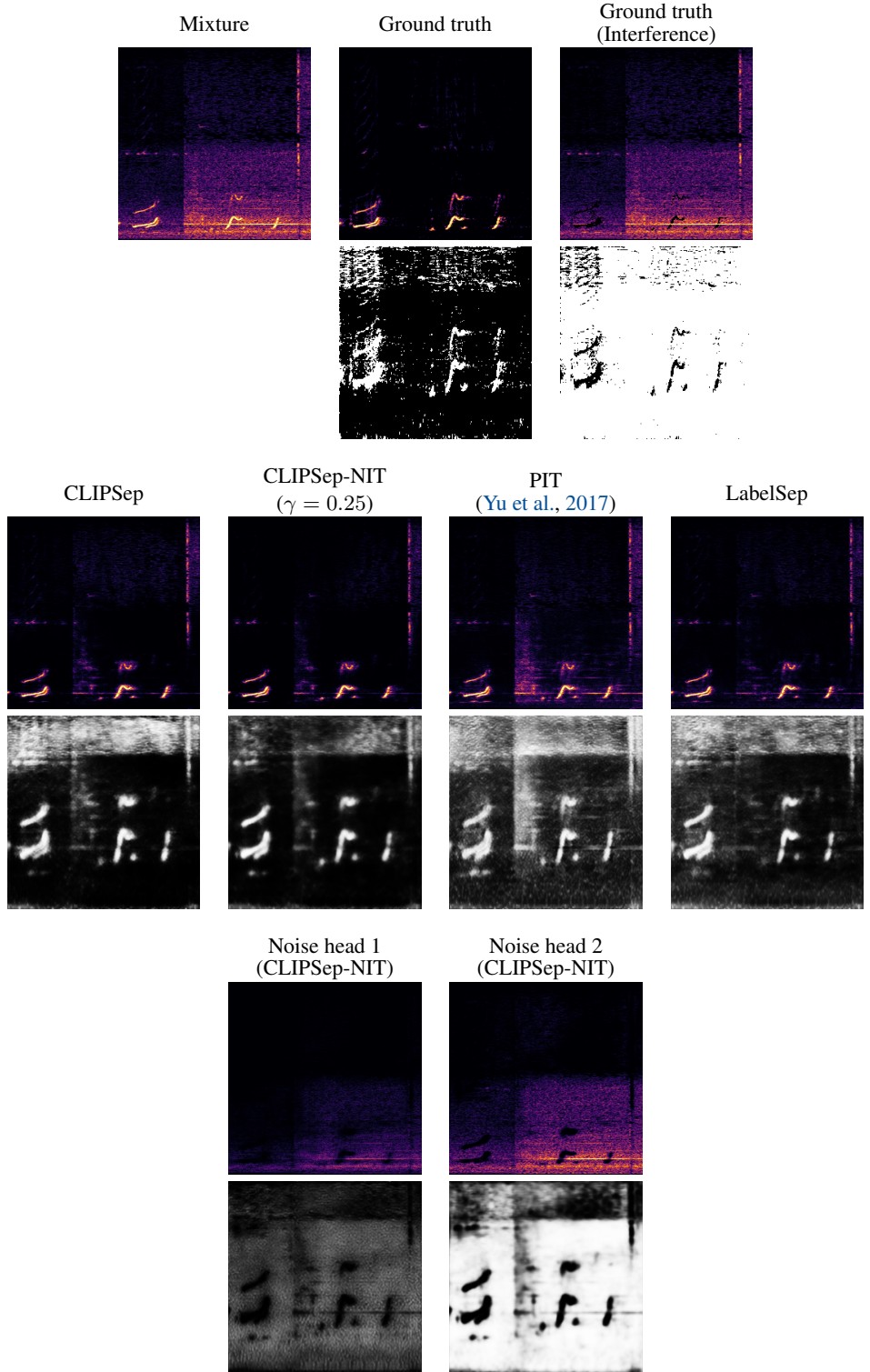

Figure 14: Example results on the VGGSound-Clean[+] dataset. Target source—"cat growling"; interference—"railroad car train wagon"; query—"*cat growling*". The spectrograms and masks are shown in the log and linear frequency scales, respectively.

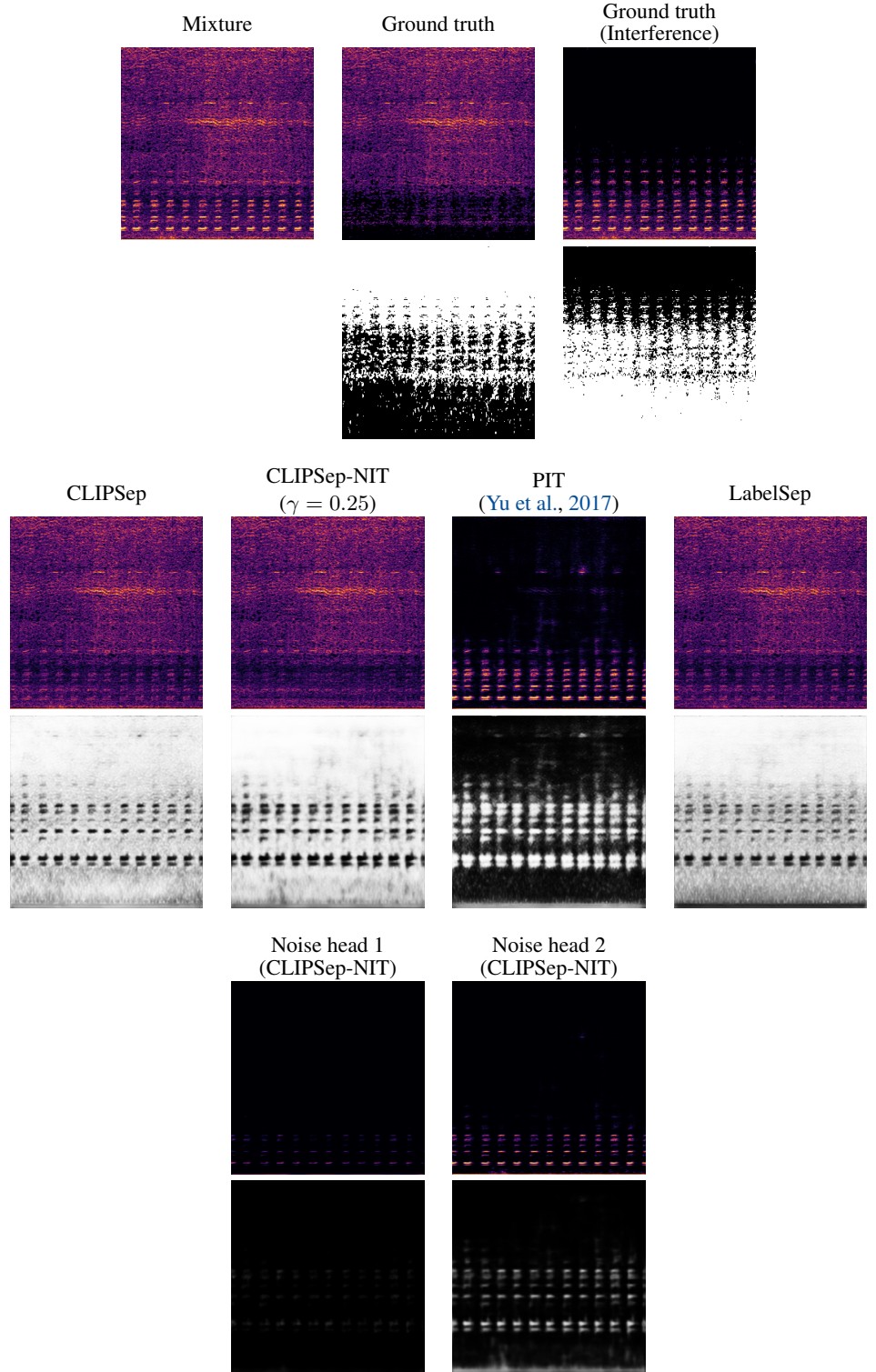

Figure 15: Example results on the VGGSound-Clean+ dataset. Target source—"electric grinder grinding"; interference—"vehicle horn car horn honking"; query—*"electric grinder grinding"*. The spectrograms and masks are shown in the log and linear frequency scales, respectively. Note that the PIT model requires a post-selection step to get the correct source. Without the post-selection step, the PIT model return the right source in only a 50% chance.

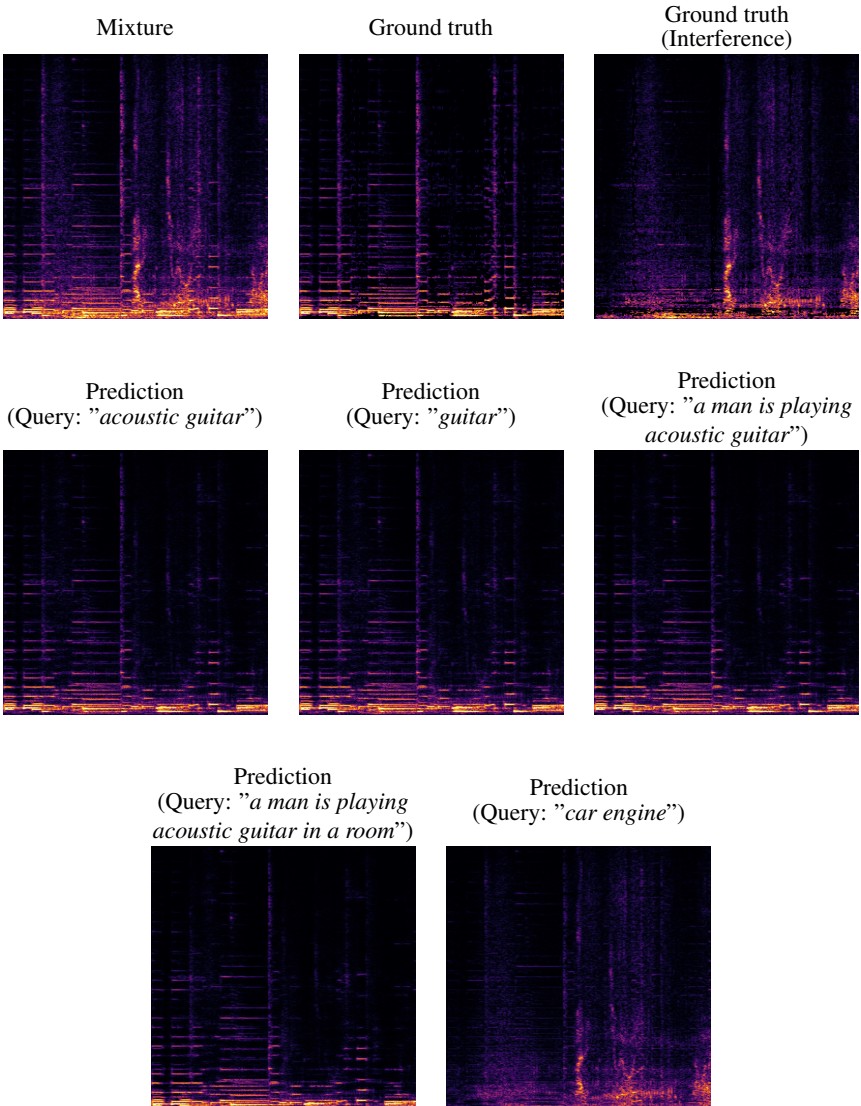

Figure 16: Query robustness experiment on the MUSIC$^+$ dataset. Target source—"acoustic guitar"; interference—"cheetah chirrup". The spectrograms are shown in the log frequency scale.

