# OpenReview forum: "CLIPSep: Learning Text-queried Sound Separation with Noisy Unlabeled Videos"
_ICLR.cc/2023/Conference — ICLR 2023 poster_

### Official Review · Reviewer_CpEt · 2022-10-21

**Confidence:** 3
**Correctness:** 4
**Technical Novelty And Significance:** 3
**Empirical Novelty And Significance:** 3
**Recommendation:** 8

**Clarity, Quality, Novelty And Reproducibility:**

Novelty: This is the first work to show training a text-queryable sound separation model trained on unlabeled video data.

Reproducibility: All code and pretrained models will be made available.

Overall clarity is good, but I  have a few suggestions:

- Section 2.3: My understanding is that the CLIP model is used as is without any training or finetuning. I think the final sentence of this paragraph could be reworded to make it clear that the part of the model you're optimizing doesn't include CLIP.

- The paper mentions a few times that the model and code is based on Sound-of-Pixels. I realize that the techniques in this paper are different than the SOP approach, but I think it would be helpful to have those differences called out explicitly because important parts are reused.

- For the architecture, I'd like to hear more about the intuition behind having the U-Net output k masks without any conditioning on the separation query. Rather than having the query vectors mix the intermediate masks, why not just condition mask generation on the query?

- Why are the noise heads discarded at test time? Is the intuition that you're training the U-Net to use some of its k masks to specialize in noise and then not be utilized by the query vectors?

**Strength And Weaknesses:**

Strengths:

- This model shows a path toward training a sound separation model that is text queryable for arbitrary sources and can be trained on unlabeled video data.
- Results are competitive with labeled approaches in some settings.

Weaknesses:

- The goal of this approach is to be able to scale up training on an arbitrary number of in-the-wild videos. However, the model is trained and evaluated only on relatively small and clean datasets. Even when the data is somewhat noisy (e.g., the offscreen noises in VGGSound), the model starts to exhibit difficulties using only text queries. The authors acknowledge these issues in the Discussion section and provide some ideas for improvement, but I'm concerned that we don't know how well the model will actually scale up to in-the-wild video datasets. It's possible that entirely different techniques will end up being needed to get to that level of unlabeled training.
  - *Update*: After discussion with the authors, I realized I misunderstood the scale of VGGSound and how representative it is of "in the wild" audio, so I am much less concerned with how well this technique will scale up.

- Motivation for some of the architecture design choices is not fully explained and alternatives are not fully explored (details below).
  - *Update*: After discussion with the authors, they have updated the paper to explain some of these choices. I found the discussion around "early fusion" vs. "late fusion" particularly interesting.

**Summary Of The Paper:**

CLIPSep demonstrates how a pretrained CLIP model can be used to train a source separation model using unlabeled videos and achieve competitive results in some settings.

**Summary Of The Review:**

CLIPSep shows a novel approach to training source separation on unlabeled videos. However, I am concerned that the main value to this approach will come from scaling up the training dataset to many in-the-wild videos, but that setting was not attempted in this paper. As shown by the VGGSound results, it's possible that there will be problems with the noisiness of in-the-wild videos that prevent this technique from working without additional insights and modifications.

*Update*: As mentioned above, I am now much less concerned about the ability of this technique to scale up.

---

> ### Author Response · Authors · 2022-11-15
> **Response to Reviewer CpEt  part 1/2**
>
> We would like to thank the reviewer for the thoughtful comments and  valuable feedback on how to further improve our paper. We are encouraged that the reviewer finds the proposed method novel and the paper clearly written. We have worked towards addressing all these comments, updating the manuscript accordingly, as well as addressing the reviewer’s concerns in this response.
>
> > However, the model is trained and evaluated only on relatively small and clean datasets.
>
> In section 4.2, we trained and evaluated our model on the VGGSound dataset, a large and noisy dataset. The VGGSound dataset contains 190k 10-second in-the-wild videos on YouTube, where the audio is rather noisy and often contains offscreen sounds. We note that it is 45 times larger than the dataset used in (Ilya Kavalerov et al., Universal sound separation, 2019).
>
> > Even when the data is somewhat noisy (e.g., the offscreen noises in VGGSound), the model starts to exhibit difficulties using only text queries.
>
> We could not identify the source of this claim. In Table 3, we show that the CLIPSep-NIT trained on noisy data performs competitively with the fully supervised model on noisy evaluation set, MUSIC+. Please note that only the target sound must be clean for correct SDR calculation, and interference sounds are noisy, containing off-screen sounds. We also provide the demo on our anonymous demo website, where the proposed model is able to separate the sound queried by the text in noisy, in-the-wild videos. (https://dezimynona.github.io/separation/)
> The discussion in Section 5 is regarding the requirement on a certain level of audio-visual correspondence in training data, which does not need to be perfect, as we showed in Section 4.2. We note that such audio-visual correspondence issues can be solved by self-supervised audio-visual correspondence prediction (Arandjelovic & Zisserman, 2017a;b).
>
> >Section 2.3: My understanding is that the CLIP model is used as is without any training or finetuning. I think the final sentence of this paragraph could be reworded to make it clear that the part of the model you're optimizing doesn't include CLIP.
>
> Yes, your understanding is correct – the CLIP model is frozen. We thank the reviewer for the suggestion. We have rephrased the sentence to clarify this point.
>
> >The paper mentions a few times that the model and code is based on Sound-of-Pixels. I realize that the techniques in this paper are different than the SOP approach, but I think it would be helpful to have those differences called out explicitly because important parts are reused.
>
> The difference between CLIPSep and Sound-of-Pixels lies in the query model shown in Figure 2. We used a frozen pretrained CLIP model and a learnable projection layer while Sound-of-Pixels used a learnable ResNet-based network (they called the model as Video Analysis Network). Other part of network architecture and loss function are the same. We clarified this in Section 3.1. The CLIPSep-NIT extends the CLIPSep by adding the noise heads.
>
> >For the architecture, I'd like to hear more about the intuition behind having the U-Net output k masks without any conditioning on the separation query. Rather than having the query vectors mix the intermediate masks, why not just condition mask generation on the query?
>
> We thank the reviewer for the thoughtful comment. Actually, we did try conditioning the audio model with the query vectors and let the audio model directly predict the mask, which we call “early fusion”. We applied this modification for both SOP and CLIPSep models, however, we observe that this architecture is prone to overfitting. We hypothesize that this is because the audio model is powerful enough to remember the subtle clues in the query vector, which hinder the generalization to a new sound and query. In contrast, the proposed architecture first predicts overdetermined masks and then combines them on the basis of the query vector. This “late fusion” architecture is shown to generalize well, probably because the audio network is powerful enough to predict intermediate masks for all sources and the linear combination of them requires only few parameters, which avoids overfitting. We have added an explanation in the Appendix C.

---

> > ### Comment · Reviewer_CpEt · 2022-11-15
> > **Response**
> >
> > Thanks for your response and clarifications, I found them very helpful. In particular:
> >
> > - I think I misunderstood the scale of VGGSound and how representative it is of "in the wild" audio. This is my fault because the paper does describe it as such, and I apologize for that. I do think mentioning that the data is sourced from YouTube might prevent similar misunderstandings in the future. With this correct understanding, I'm now much less concerned about the model's ability to scale up.
> >
> > - The discussion added about the "early fusion" vs. "late fusion" models is fascinating! If there is room, I'd suggest trying to fit some of that intuition into the main body of the paper.

---

> > > ### Author Response · Authors · 2022-11-21
> > > **Response**
> > >
> > > We are glad that the reviewer finds our response helpful. We thank the reviewer for reconsidering the rating and providing additional feedback on the discussion about the model architecture. We will try to fit the intuition into the main body of the final paper with a pointer to the discussion in the Appendix.

---

> ### Author Response · Authors · 2022-11-15
> **Response to reviewer CpEt part 2/2**
>
> >Why are the noise heads discarded at test time? Is the intuition that you're training the U-Net to use some of its k masks to specialize in noise and then not be utilized by the query vectors?
>
> We discarded the noise head because we expect the noise heads to contain only noise, which is irrelevant to the query. The noise heads are introduced to enable the training on noisy targets. For example, we want to train the model to separate the sound ‘A’, but we only have access to the noisy data ‘A+N’, where N is the noise. When we train CLIPSep on the noisy data, CLIPSep learns to output ‘A+N’ because the target signal is ‘A+N’ and CLIPSep has one head for each query. In contrast, as CLIPSep-NIT has noise heads, it is possible to assign ‘A’ to the query head and ‘N’ to the noise head, yet, we can train the model with the noisy target ‘A+N’ via the proposed noise invariant training. At test time, since our goal is to separate not ‘A+N’ but ‘A’, we discarded the noise heads.
>
> >However, I am concerned that the main value to this approach will come from scaling up the training dataset to many in-the-wild videos, but that setting was not attempted in this paper. As shown by the VGGSound results, it's possible that there will be problems with the noisiness of in-the-wild videos that prevent this technique from working without additional insights and modifications.
>
> As discussed above, we trained and evaluated the model on large, in-the-wild video datasets, which contains noise and offscreen sounds. To address the noisiness of in-the-wild videos, we proposed the noise invariant training and showed the effectiveness in Section 4.2. The discussion in Section 5 is about the consideration required for further scaling up the size of the dataset.

---

### Official Review · Reviewer_8zRs · 2022-10-24

**Confidence:** 4
**Correctness:** 3
**Technical Novelty And Significance:** 3
**Empirical Novelty And Significance:** 3
**Recommendation:** 8

**Clarity, Quality, Novelty And Reproducibility:**

The paper reads well in general. In terms of novelty, due to the fact that this paper proposes a new training methodology which enables training with audio-video pairs, it seems to differentiate itself from the existing papers.

**Strength And Weaknesses:**

Strengths:
- The proposed problem is definitely interesting, and I can see the practical applications of this system.
- The results (shared in the link https://dezimynona.github.io/separation/) seems to suggest that the system is doing what is intended.

Weaknesses:
- I think it would have been nice to also compare with a baseline system which uses sentence embeddings as a guide. This paper could be a nice point of comparison https://arxiv.org/pdf/2203.15147.pdf. You could have done this comparison in two ways. 1) On your experiments you can directly train this model and compare 2) You could have taken a pretrained systems for both your approach, and the baseline and compare in a zero-shot manner. The VGGSound+None experiment that you have on your demo page is a nice option for this.
- There is little difference between the separation quality of Clipsep and Clipsep+NIT. In some of the examples on your demo page the two methods sound very similar.

**Summary Of The Paper:**

This paper proposes a source separation system for text-queried source separation. The authors propose to train the system with a picture query during training time, however in inference time they use text for the query. In addition to the basic system, they also propose to add a mixit layer at the end of the pipeline to increase the noise robustness of the system.

**Summary Of The Review:**

I think this paper proposes an interesting training methodology. I think it's above the acceptance threshold. My only problem with it is the lack of comparison with text-query-only models. (See my comment above)


---------------


Update after rebuttal: The authors provided a BERT based baseline, and I increased my score.

---

> ### Author Response · Authors · 2022-11-15
> **Response to reviewer 8zRs**
>
> We appreciate the reviewer for the thoughtful review and constructive comments. We are encouraged that the reviewer finds our approach novel and interesting. Please find our response below.
>
> > My only problem with it is the lack of comparison with text-query-only models.
>
> One of the text-query-only models we compare with is the CLIPSep-TEXT model, a variant of the proposed model. As the CLIP model is well known to capture the semantic structure of language [1,2], training and testing the model with CLIP-text embedding serves as a strong upper baseline. We emphasize that CLIPSep-TEXT is a supervised model as it requires labels for training.
> [1] Ramesh et al., Hierarchical Text-Conditional Image Generation with CLIP Latents (https://cdn.openai.com/papers/dall-e-2.pdf)
> [2] Wu et al., Wav2CLIP: Learning Robust Audio Representations From CLIP, ICASSP 2022 (https://arxiv.org/pdf/2110.11499.pdf)
>
> >I think it would have been nice to also compare with a baseline system which uses sentence embeddings as a guide. This paper could be a nice point of comparison https://arxiv.org/pdf/2203.15147.pdf. You could have done this comparison in two ways ~
>
> We thank the reviewer for the suggestion. Unfortunately, the code of the paper (Liu et al., 2022) has not been released yet, so we could not reproduce the model at the time of submission. However, as discussed above, we believe that CLIPSep-TEXT serves as a strong supervised baseline method since it uses labels for training similar to (Liu et al., 2022), where they extracted a query vector using BERT while we used the CLIP-text embedding.
>
> >There is little difference between the separation quality of Clipsep and Clipsep+NIT. In some of the examples on your demo page the two methods sound very similar.
>
> We welcome the reviewer’s comment. To evaluate whether the SDR results aligned with perceptual quality, we have conducted an additional subjective test. As done in the Sound of Pixel paper (Zhao et al., 2018), audio samples are randomly presented to evaluators, and the following question is asked: “Which sound do you hear? 1. A, 2. B, 3. Both, or 4. None of them”. Here A and B are replaced by labels of their mixture sources, e.g. A=”accordion”, B=”engine accelerating”. Ten samples are evaluated for each model and 16evaluators have so far participated in the evaluation (We will try our best to recruit more participants by the end of discussion Stage 1). We have presented the results in Appendix F. The results indicate that the evaluators more often choose the correct sound source for CLIPSep-NIT (83.8%) than CLIPSep (66.3%). We believe that these results and the SDR values indicate the effectiveness of the proposed NIT both quantitatively and qualitatively.

---

> > ### Comment · Reviewer_8zRs · 2022-11-30
> > **Thanks for your response**
> >
> > I haven't worked with the CLIP model before myself, but my intuition is that it's multimodal model for text-images. Therefore it would have been nice to compare with a system that directly uses text-embeddings. (to potentially establish an upper bound as what you do is a solution to the case where you do not have text, but have the video). It could be something like using BERT in your pipeline, and measure the performance.
> >
> > In any case, I think the paper has its merits, and I am keeping the same score I gave before. (6)

---

> > > ### Author Response · Authors · 2022-12-03
> > > **Additional baseline results with BERT model**
> > >
> > > We thank the reviewer for the additional feedback. As suggested by the reviewer, we have trained a model with BERT embeddings by replacing the CLIP model in CLIPSep with a BERT encoder, which we call BERTSep. Following Liu et al. 2022, we use the pretrained BERT model with 4 Transformer encoder blocks, each with 4 attention heads and 256 dimensions. We trained BERTSep on the VGGSound dataset using the labels provided in the dataset, where the BERT model is frozen.
> > >
> > > We summarize the results as follows.
> > >
> > > + Results on the VGGSound+ dataset
> > > |Model | Mean SDR | Median SDR |
> > > |:-|:-:|:-:|
> > > | BERTSep (new baseline) | 5.09 $\pm$ 0.80 | 5.49 |
> > > | CLIPSep-Text | 5.49 $\pm$ 0.82 | 5.06 |
> > >
> > > + Results on the VGGSound-Clean+ dataset
> > > | Model | Mean SDR | Median SDR |
> > > |:-|:-:|:-:|
> > > | BERTSep (new baseline) | 4.67 $\pm$ 0.44 | 4.41 |
> > > | CLIPSep-Text | 10.73 $\pm$ 0.99 | 9.93 |
> > >
> > > Here are some observations:
> > >
> > > + __BERTSep performs similarly CLIPSep-Text on the VGGSound-Clean+ dataset__. This is expected as both BERTSep and CLIPSep-Text use query vectors obtained from the same texts and the query vectors are seen during training. (Please note that CLIPSep-Text does _not_ use image for either training or testing, but rather it uses text only as in BERTSep.)
> > > + However, __the performance of BERTSep is significantly lower than that of CLIPSep-Text on the MUISC+ dataset__. This indicates that BERTSep does not generalize to unseen text queries in MUSIC dataset. We hypothesize that CLIP text embeddings capture the similarity of timbre better than BERT embeddings. For example, we expect the ‘viola’ sound more similar to ‘violin’ than ‘flute’. As the CLIP model is pretrained on language-image contrastive learning, it is aware of the visual similarity between the violin and viola (i.e., the viola and violin look similar to each other; but the flute looks different from the viola). Hence, the CLIP text embeddings of ‘viola’ and ‘violin’ can be closer than that of ‘flute’. In contrast, BERT is trained on text data only, and thus it would be more difficult to learn that ‘viola’ should be semantically closer to ‘violin’ than ‘flute’.

---

> > > > ### Comment · Reviewer_8zRs · 2022-12-03
> > > > **Text encoder**
> > > >
> > > > Great, I thank the authors for the additional baseline. I increase my score to 8.

---

### Official Review · Reviewer_dcmL · 2022-10-24

**Confidence:** 3
**Clarity, Quality, Novelty And Reproducibility:** 1. Clarity
- What's up with the Figur…
**Correctness:** 3
**Technical Novelty And Significance:** 3
**Empirical Novelty And Significance:** Not applicable
**Recommendation:** 8

**Strength And Weaknesses:**

Strengths:
1. The main strength is that the method is novel. I like this idea a lot and think there's something materially interesting if you ramp up the dataset size.
2. The comparisons are also clear. The tables show the delineations between the models that you compare and I don't have trouble understanding what's going on wrt numbers.


Weaknesses:
1. The explanation of the model feels like some info is left out, notably from where the images are extracted with respect to the audio. As I understand, there is a singular image per video (2 total to be exact), but it's unclear how the audio is determined around that. It can't be instantaneous. Is it 10 seconds around it? Maybe I'm missing it, but this seems important for reproduction.
2. There should be audio samples here. It's hard to truly evaluate what's going on without audio samples. I don't see any such links in the paper.
3. I don't understand at all what is section 4.1. What is the task? I read through it a few times and it's unclear to me what you're actually doing there.

**Summary Of The Paper:**

The paper describes a self-supervised way to do sound separation using a frozen pre-trained CLIP, along with video data (assumed to also have audio).

The core method of CLIPSep is shown in Fig 2. During training, they run the frames of two different videos through CLIP in order to independently get embeddings, which are then projected into another space by a learnable projection mapping. In parallel, they add together the audio streams of both videos, encode this as a spectrogram, and then run that through an Audio UNet. They then independently combine the output of the UNet with each video's projections in order to predict an audio mask. That audio mask is compared against the true audio mask for the video in order to get a loss.

Figure 3 expands on CLIPSep and introduces CLIPSep-NIT in order to better account for noisy streams of audio. It's more complicated, but the gist is to create audio masks that account for the noise found in in-the-wild videos. This is patterned after the MixIT approach from Wisdom et al.

They then show that this self-supervised approach can be comparable to supervised datasets on two different tasks involving mixing test VGGSound and eval MUSIC+ with VGGSound.

**Summary Of The Review:**

Before I increase my score, I would want to see the paper improve significantly wrt clarity, notably section 4.1 and Figure 3. I would also like to see a better explanation for the evaluation approach in 4.2 and perhaps something to support it elsewhere in the literature. If those are satisfied I will increase my score because I do like this paper and think the underlying method deserves to be recognized.

---

> ### Author Response · Authors · 2022-11-15
> **Response to reviewer dcmL**
>
> We appreciate the reviewer for the thoughtful review and constructive feedback. We are encouraged that the reviewer finds our approach novel and interesting, the results are convincing, and the paper clear. We believe our approach provides a new paradigm to approach unsupervised text-driven universal sound separation. We respond to the reviewer’s comments below.
>
> >The explanation of the model feels like some info is left out, notably from where the images are extracted with respect to the audio. As I understand, there is a singular image per video (2 total to be exact), but it's unclear how the audio is determined around that. It can't be instantaneous. Is it 10 seconds around it? Maybe I'm missing it, but this seems important for reproduction.
>
> >There should be more details about the image + audio pairings. I see in the Appendix that they use 4 second audio clips, but where is the image drawn from?
>
> Audio and images are time-aligned. During training, we randomly sampled a center frame from a video and extracted three frames (images) with 1-sec intervals and 4-sec audio around the center frame. During inference, for image-queried models, we extracted three frames with 1-sec intervals around the center of the test clip. We have clarified this in Appendix C.
>
> >There should be audio samples here. It's hard to truly evaluate what's going on without audio samples. I don't see any such links in the paper.
>
> We provided the audio samples on our anonymous demo website (https://dezimynona.github.io/separation/), as mentioned in Section 1 and 4.2. Please refer to the demo website. We have also added the link in Section 7.
>
> >I don't understand at all what is section 4.1. What is the task? I read through it a few times and it's unclear to me what you're actually doing there.
>
> The task is sound separation. We mix two sounds from the MUSIC dataset and evaluate the separation quality, as done in Sound of Pixel (Zhao et al. 2018). This experiment on the clean dataset enables us to focus on evaluating the quality of CLIP-based query vectors and the zero-shot modality transferability in an ideal case (without noise in training data). We have clarified the task description in Section 4.1.
>
> >What's up with the Figure 3 graphic? The clarity of this paper would be helped a lot if you made the 2nd half of this better because it's hard to grok what's going on in the text itself. ~
>
> We apologize for the lack of clarity in Figure 3. The noise heads are grayed out because they will be discarded at test time. The dark red and blue arrows represent the interchangeable loss computation paths. We have updated the figure and caption to clarify them.
>
> > it does seem strange that they did this mixing of datasets. Is that what other papers are doing? I'm not as familiar w this field as I'd like to be to question that, but it is does seem kind of strange.
>
> Yes, this is the standard practice for evaluating a source separation system. The signal-to-distortion ratio (SDR) is a commonly used evaluation metric for source separation [1], [2] (note that there are some variants such as SI-SNR, Normalized-SDR, etc.). Since the calculation of these metrics requires the ground truth source, it is a common practice to evaluate a source separation model by artificially mixing clean sources and comparing the separations with the ground truth sources.
> In Section 4.2, we aim at evaluating the proposed model on noisy data, namely, VGGSound. As discussed at the beginning of Section 4.2, the target source must be clean while interference can be noisy. Thus, we consider the two settings, MUSIC(clean)+VGGSound(noisy) and VGGSound-Clean+VGGSound(noisy).
> [1] The 2018 Signal Separation Evaluation Campaign (https://arxiv.org/abs/1804.06267)
> [2] SDR - half-baked or well done? (https://arxiv.org/abs/1811.02508)

---

> > ### Comment · Reviewer_dcmL · 2022-12-01
> > **Response**
> >
> > Acknowledged. I'll increase my score to 7.

---

### Official Review · Reviewer_vU71 · 2022-10-25

**Confidence:** 4
**Correctness:** 3
**Technical Novelty And Significance:** 3
**Empirical Novelty And Significance:** 3
**Recommendation:** 8

**Clarity, Quality, Novelty And Reproducibility:**

The clarity and quality is good and generally well written. It lacks a certain level of final polish to make 1. how it differs previous, comparable work and 2. the findings absolutely clear. Most of the details can be found in the text, but summaries and figures could make it more obvious. For example Figure 4, showing mean SDR for image and text inputs in test, for models training with different modalities. This would be clearer in a table, ie
                            |   Test Modality              |
     Train Modality |   Image  |  Text  |   Both |
ClipSep (Image)   |    7.5     |   5.5    |  ?    |
ClipSep (Text)      |    6.2     |   8.1    |  ?    |
ClipSep (Both)     |    8.1     |   8.2    |    ?    |
* #s are approximately estimated from figure 4.
Here one can see how good the model is if the train/test modalities are matched. There's more lost when trained on image and tested on text (unfortunately the main goal of the paper). Using both in train help significantly. Could you test with both? Would be an interesting result.

The paper is novel in a narrow sense, since the field has a lot of work in audio separation via query and addressing unsupervised separation of audio sources.
The unsupervised separation of audio by query is similar to the work in:
* Liu et al., Separate What You Describe: Language-Queried Audio Source Separation, Proc Interspeech 2022
  - text queries are used to select a source to separate in audio-only samples
  - the paper under review has the addition of a visual modality to improve the correspondence between text and the input modes.
* Zhao et al. The Sounds of Pixels. ECCV 2108 (cited by paper and base implementation)
  - unsupervised audio-visual source separation in videos with musicians playing music, selection/query by image~
  - the paper under review adds a text query component to select the source to separate out, and a Noise Invariant Training scheme to cope with (audio) noise sources that have no correspondence in the video. it also focuses on unconstrained sound vs only music in Zhao.
* Wisdom et al. Unsupervised Sound Separation Using Mixture Invariant Training
  - unsupervised audio separation, mixture of mixtures invariant training
  - doesn't provide a means to select a single source to extract (separates all sources)

The paper uses publicly presented data sources and published github repositories. The paper should be relatively easy to reproduce.

Minor comments
- are the masks used in the paper binary or ratio? Zhao mentions that both are possible.
- 4th line in Conclusion has a typo "language pre*training*".

**Details Of Ethics Concerns:**

I am unsure of this, but I believe that this work used YouTube videos as training data and thus requires downloading them which is against YouTube's term of service. There has been a lot of published work though that has used YouTube as a data source such as AudioSet [1] and VoxCeleb [2]. [1] is even from Google, YouTube's parent.

**Strength And Weaknesses:**

Strengths:
* A new configuration of querying by text to separate out an audio source in a video with sources that have corresponding audio and visual signals.
* Shows performance competitive with state-of-the-art in sound separation

Weaknesses
* Tests are made by artificially combining samples of YouTube videos. Can you conduct test results on naturally occurring mixtures?
* Results report an automatically computed quantitative metric, ie SDR. It is unclear whether how this corresponds to actual user preferences. Since the results are close, could a qualitative survey be conducting comparing the results of PIT with CLIPSep, similar to how they were done in Sound of Pixels using Mechanical Turk?



**Summary Of The Paper:**

The paper under review proposes a method of selecting a single sound source from a mixture of sounds in a video via a text description of the visual component of the video. The system can be trained on unlabeled data, aka unsupervised training. This is a novel configuration of using a pre-trained audio-visual correspondence model to allow text queries to select the single audio source to separate from a mixture in the video. Unlike what is claimed in the paper though in section 4.1, work was published this year on querying by text to separate a source from an audio mixture (this is understandable given timing). There is also a contribution of a form of noise invariant training that allows for the model to account for sounds in the mixture that have no correspondence in the video. The results are conducted on test sets, MUSIC and VGGSound-Clean, that have audio collected from the wild (YouTube), however they have been artificially mixed to yield multiple sound sources. The results are competitive with PIT, although PIT has a "post-processing" requirement.



**Summary Of The Review:**

Overall the paper is novel in a narrow sense. It builds on the Sound Of Pixels work but adding a method of textual query. The results are good, demonstrating the approach is viable, however in the opinion of this reviewer, not overwhelming excellent (other reviewers may disagree). It feels more incremental than ground breaking, hence the recommendation to marginally accept.

---

> ### Author Response · Authors · 2022-11-15
> **Response to reviwer vU71 part 1/2**
>
> We would like to thank the reviewer for their valuable feedback on how to further improve our paper. We have extensively worked towards addressing all these comments by conducting additional subjective experiments, updating the manuscript accordingly, as well as addressing the reviewer’s concerns in this response.
> To our knowledge, our proposed method is the first work on self-supervised text-queried sound separation, which liberates us from the need of costly human-annotated audio-text pair data and enables the learning of  text-queried sound separation using unlabeled video.
>
> >Tests are made by artificially combining samples of YouTube videos. Can you conduct test results on naturally occurring mixtures?
>
> We understand the reviewer’s concern. However, since the calculation of SDR requires the ground truth signal and cannot be calculated for natural mixture, it is a common practice to use artificial mixtures for evaluating the separation quality with the objective metrics, SDR. In the speech domain, some work proposed to use neural networks that predicts mean opinion score (MOS) of human-evaluations. However, evaluating the separation of naturally occurring “universal” sound mixtures quantitatively without access to the ground truth audio still remains an open question. We provide audio samples of separating naturally occurring mixtures in our anonymous demo page (in the “VGGSound+None” section) for the readers to assess the perceptual quality.
>
> >Results report an automatically computed quantitative metric, ie SDR. It is unclear whether how this corresponds to actual user preferences. Since the results are close, could a qualitative survey be conducting comparing the results of PIT with CLIPSep, similar to how they were done in Sound of Pixels using Mechanical Turk?
>
> We thank the reviewer for the suggestion. We have conducted an additional subjective listening test to evaluate whether the SDR results aligned with perceptual quality. As done in the Sound of Pixel paper (Zhao et al., 2018), audio samples are randomly presented to evaluators, and the following question is asked: “Which sound do you hear? 1. A, 2. B, 3. Both, or 4. None of them”. Here A and B are replaced by labels of their mixture sources, e.g. A=”accordion”, B=”engine accelerating”. Ten samples (including naturally occurring mixture) are evaluated for each model and 16 evaluators have so far participated in the evaluation (We will try our best to recruit more participants by the end of discussion Stage 1). We have presented the results in Appendix F. The results indicate that the evaluators more often choose the correct sound source for CLIPSep-NIT (83.8%) than CLIPSep (66.3%). We believe that these results and the SDR values indicate the effectiveness of the proposed NIT both quantitatively and qualitatively.
>
> >Most of the details can be found in the text, but summaries and figures could make it more obvious. For example Figure 4, showing mean SDR for image and text inputs in test, for models training with different modalities.
>
> We believe there is a misunderstanding regarding Figure 4 and the CLIPSep-Hybrid due to the lack of clarity. The values in Figure 4 are all present in Table 2 (i.e., Figure 4 is a subset of Table 2), and the CLIPSep-Hybrid does not concatenate image- and text-CLIP embeddings but use each in alternation during training (we alternatively feed either image- or text-embedding during training, as described in Appendix C). We included Figure 4 for directly comparing the performance within the test-mode (indicated by the legend) and among the training mode (x-axis). We have clarified the explanation in Section 4.1.

---

> ### Author Response · Authors · 2022-11-15
> **Response to reviwer vU71 part 2/2**
>
> >The paper is novel in a narrow sense, since the field has a lot of work in audio separation via query and addressing unsupervised separation of audio sources. The unsupervised separation of audio by query is similar to the work in: ~
>
> We thank the reviewer for pointing out the three highly-relevant papers we discussed in Section 2. However, we want to emphasize that it is not straightforward to move from an image-queried model to a text-queried model because previous text-queried separation models (including the study by Liu et al. (2022))  require considerable effort for collecting a large amount of manually annotated (or captioned) data for training. Indeed, Liu et al. (2022) used human-annotated text-audio pair data and screened the audio from AudioCap dataset using tag information.  In this sense, their approach is not an unsupervised method. Our proposed method liberates us from the need of costly human-annotated audio-text pair data and enables the learning of  text-queried sound separation using unlabeled video. To our knowledge, this represents the first work on self-supervised text-queried sound separation.
> Our method has also shown the potential of the idea of leveraging pretrained multimodal models (e.g., images and text; audio and images) for bridging modalities that are less frequently observed together (e.g., audio and text).
>
> >The paper uses publicly presented data sources and published github repositories. The paper should be relatively easy to reproduce.
>
> As described in Section 1 and 7, we released inference code and will release all the remaining code including data preprocessing for reproducibility.
>
> >are the masks used in the paper binary or ratio? Zhao mentions that both are possible.
> We used the ideal binary masks as the target for training, and used real-valued mask predictions (the sigmoid outputs) used for inference. We described this in Appendix C.
>
> >4th line in Conclusion has a typo "language pretraining".
>
>  We thank the reviewer for pointing this out. We have corrected the typo.
>
> > Overall the paper is novel in a narrow sense. It builds on the Sound Of Pixels work but adding a method of textual query. The results are good, demonstrating the approach is viable, however in the opinion of this reviewer, not overwhelming excellent (other reviewers may disagree). It feels more incremental than ground breaking, hence the recommendation to marginally accept.
>
> Provided the proposed method (CLIPSep-NIT) does not use any labeled data, we believe that achieving competitive results as the fully supervised method (CLIPSep-Text) on MUSIC+ dataset is surprisingly good. On the other hand, we agree that there is room for improvement on the VGGSound-Clean+ dataset. We will continue to work on scaling up the dataset, as discussed in Section 5.
> We also would like to emphasize that, to our knowledge, this is the first work on self-supervised text-queried sound separation, which does not require costly human-annotated audio-text pair data.
>
> >I am unsure of this, but I believe that this work used YouTube videos as training data and thus requires downloading them which is against YouTube's term of service.
>
> We only use publicly available datasets, VGGSound and MUSIC, which are already used in many other publications.

---

> > ### Comment · Reviewer_vU71 · 2022-12-03
> > **Response to authors**
> >
> > > We only use publicly available datasets, VGGSound and MUSIC, which are already used in many other publications.
> >
> > Although YouTube videos are publicly viewable, and although many publications have used them as data for research, in general I believe downloading them is against the terms of service.
> >
> > > As described in Section 1 and 7, we released inference code and will release all the remaining code including data preprocessing for reproducibility.
> > My original comment was noting that this is indeed a good thing and much appreciated.
> >
> > > We provide audio samples of separating naturally occurring mixtures in our anonymous demo page (in the “VGGSound+None” section) for the readers to assess the perceptual quality.
> > > The results indicate that the evaluators more often choose the correct sound source for CLIPSep-NIT (83.8%) than CLIPSep (66.3%). We believe that these results and the SDR values indicate the effectiveness of the proposed NIT both quantitatively and qualitatively.
> >
> > Thanks for conducting these results, it is much appreciated. Given the quantitative and qualitative results and the clarifications by the authors, I can increase my score to 8.

---

### Official Review · Reviewer_LZWB · 2022-10-26

**Confidence:** 4
**Correctness:** 4
**Technical Novelty And Significance:** 3
**Empirical Novelty And Significance:** 3
**Recommendation:** 6

**Clarity, Quality, Novelty And Reproducibility:**

Clarity: the paper is very clear. I only have minor suggestions for improvement (see weaknesses)

Quality: high quality. Evaluation is solid and compares to relevant baselines. Some nice additional information is provided in the appendices.

Novelty: paper is novel, in that it proposes a text-driven separation method that can be trained on noisy data, and minor novelty in the noise invariant training.

Reproducibility: the code and models are made available.

**Strength And Weaknesses:**

### Strengths

1. To my knowledge, this is the first method to train text-queried separation on noisy mixtures.

2. The evaluation is done on both MUSIC+ and VGGSound-Clean+, measuring performance on both music separation and universal separation, and these results are convincing.

3. Paper includes link to anonymized demo page, which is convincing.

### Weaknesses

1. I think the paper makes the post-selection step required for a MixIT model to be harder than it actually is. For a MixIT-trained model with N outputs, it's pretty easy to pick a source, e.g. with a sound classification network. This setup was actually proposed with a classification-regularized loss in:
Wisdom, Scott, Aren Jansen, Ron J. Weiss, Hakan Erdogan, and John R. Hershey. "Sparse, efficient, and semantic mixture invariant training: Taming in-the-wild unsupervised sound separation." In 2021 IEEE Workshop on Applications of Signal Processing to Audio and Acoustics (WASPAA), pp. 51-55. IEEE, 2021. (https://arxiv.org/pdf/2106.00847.pdf)
Another advantage of MixIT is that the outputs are more interpretable, compared to models that rely on conditioning, such as the one described in this paper. Thus, I think it may be good to discuss the pros and cons of separate-then-select versus conditional separation in the paper.

2. This statement is a bit incorrect:

    "However, AudioScope still requires a post-selection process if there is more than one predicted on-screen channel."

    The goal of AudioScope is to recover all on-screen sounds in a single channel, which is what the model does: it uses on-screen probabilities as mixing weights across the sources.

3. "where s1, . . . , sn are the n audio sources,": in practice, these are mixtures, right? The model is just assuming that they are single sources. it might be good to refine the terminology here a bit.

4. Some explanation of why k masks are predicted, then combined, would be good. I think this is kind of analogous to the multiple output sources in MixIT, which can be combined for a particular user interface or output goal, e.g. AudioScope combines with on-screen probabilities to get an estimate of on-screen sound.

5. The equation for computing the overall source mask from the k masks is confusing. What does the \odot versus the \cdot mean? If w_i is k-dimensional, I don't see a sum over k, since it's \odot'ed with scalar q_{ij} times \tilde{M}_j. Should this actually be w_{i,j}? Please specify how this is done.

6. The model uses mask-based losses, which, in my own experience, are often suboptimal compared to signal based losses (i.e. computed loss in time domain, backpropping through iSTFT applied to masked mixture STFT). Also, in the NIT loss, adding masks together and applying a ceil of 1 does not exactly correspond to adding signals in the time domain, because of STFT consistency. it would be interesting to try time-domain based losses for this network, and see if that provides any improvement. Also, the architecture in the MixIT paper used mixture consistency, so that output sources sum up to the original input mixture. This might also be a useful constraint on the architecture here.

7. I think best practice for reporting units in decibels is to use only one decimal place. Humans can often not even hear 0.1 dB of difference. Thanks, by the way, for reporting std dev from the mean and median.

8. More explanation of the motivation of NIT would be very welcome. My intuition is that it helps "soak up" extra noise by providing additional output sources, but this might not be right. Please add some explicit discussion of the motivation.

### Typos and minor comments

a. "For eaxmple," -> "For example,"

**Summary Of The Paper:**

### Summary

This paper proposes a text-queried universal sound separation model that can be trained on noisy in-the-wild videos (i.e. videos that contain both on-screen and off-screen sounds). Two versions are proposed: CLIPSep and CLIPSep-NIT (CLIPSep with noise invariant training).

CLIPSep: during training, mix audio from two videos. Extract the CLIP embedding of an image frame; from the spectrogram of the audio mixture, predict k masks; predict a k-dim query vector q_i from the CLIP embedding; predict overall mask for source i using query vector q_i to combine across the k masks, with an additional k-dimensional scaling weight w_i and scalar bias b_i; audio is reconstructed using inverse STFT on masked STFT. Training loss is weighted binary cross-entropy between estimated mask and ground-truth mask (so training requires isolated source audio from on-screen-only video). During inference, CLIP embedding is computed from text (assuming this will be close to CLIP embedding of image), and just one mask is predicted for the source described by the text.

CLIPSep-NIT: same as CLIPSep, except that for each of the n sources during training, an additional "noise" mask is predicted, which is an additional query vector that combines the k predicted masks with a noise query vector. Then during training, all permutations of the noise masks added to the source masks are considered, and the permutation with the minimum error is used. It seems the purpose of the noise masks is to "soak up" sounds not related to the CLIP embedding. At test time, the noise masks are discarded.

### Contributions

1. First text-driven separation model (to my knowledge) that can be trained on noisy videos, enabled by the NIT trick.

2. NIT is a contribution, though I feel its novelty is relatively minor, since it's just a constrained version of permutation invariant training (PIT).

**Summary Of The Review:**

Overall, a nice paper that accomplishes training text-driven separation on noisy in-the-wild data. Achieves good performance compared to prior approaches, and qualitative demos are convincing.

---

> ### Author Response · Authors · 2022-11-15
> **Response to reviwer  LZWB  part 1/2**
>
> We thank the reviewer for the thoughtful review. We are encouraged that the reviewer finds the proposed method novel, the results convincing, and the paper clearly written. We believe our approach provides a new paradigm to approach unsupervised text-driven universal sound separation. We answer the reviewer’s comments below.
>
> > I think the paper makes the post-selection step required for a MixIT model to be harder than it actually is. For a MixIT-trained model with N outputs, it's pretty easy to pick a source, e.g. with a sound classification network. This setup was actually proposed with a classification-regularized loss in: Wisdom ~
>
> We thank the reviewer for pointing the related work out. We included the suggested paper and discussed the pros and cons in Section 2. The approaches differ in how they achieve unsupervised universal sound separation and provide an interface to users. They use a pre-trained sound classification network, which requires manual labels to train, and provide the predefined labels to the separated sources (separate, label, and choose), while our approach uses the CLIP model to learn to separate sound by user-specified text on unlabeled video (query and separate). It depends on the use cases and the desired accuracy as to which interface is suitable. If users require all sound sources from a mixture, separating all sources first is a natural approach. However, the pre-trained sound classifier may not cover sound types of a user’s interest and the classification accuracy may not be very accurate on the separated sources due to artifacts. On the other hand, if a user wants to separate a specific sound, the text-query-based approach provides a natural and scalable interface. It would also be interesting to combine these approaches (e.g., separate and choose on the basis of a user-query).
>
> > This statement is a bit incorrect:
> "However, AudioScope still requires a post-selection process if there is more than one predicted on-screen channel."
> The goal of AudioScope is to recover all on-screen sounds in a single channel, which is what the model does: it uses on-screen probabilities as mixing weights across the sources.
>
> We intended to convey that the separation model in AudioScope separates M sources and the on-screen classifier assigns the probability of each source being an on-screen sound. Hence, if there are multiple sound sources on the screen, AudioScope only predicts which of the separated sources is an on-screen sound, but does not predict which object on the screen corresponds to which separated source. We have removed this sentence in the revised manuscript to avoid confusions on the goals of AudioScope.
>
> >"where s1, . . . , sn are the n audio sources,": in practice, these are mixtures, right? The model is just assuming that they are single sources. it might be good to refine the terminology here a bit.
>
> We thank the reviewer for the suggestion. Yes, these are actually mixtures as audio tracks of the videos usually contain more than one source. We have refined the terminology.
>
> >Some explanation of why k masks are predicted, then combined, would be good. I think this is kind of analogous to the multiple output sources in MixIT, which can be combined for a particular user interface or output goal, e.g. AudioScope combines with on-screen probabilities to get an estimate of on-screen sound.
>
> This is the same architecture as Sound of Pixel (Zhao et al., 2018) and analogous to MixIT (Wisdom et al., 2020). The overdetermined masks are combined on the basis of a user-provided query.  In our preliminary experiments, we also tried directly predicting the final mask by conditioning the audio model on the query vector. However, this model was prone to overfitting, possibly because the audio model is powerful enough to remember the subtle clue in the query vector. Since the audio model is powerful enough, the proposed architecture first predicts the overdetermined intermediate masks for all sources and then combines the intermediate masks on the basis of the query vector, which avoids the overfitting problem due to the simple fusion step. We added Appendix C to explain this.
>
> >The equation for computing the overall source mask from the k masks is confusing. What does the \odot versus the \cdot mean? If w_i is k-dimensional, I don't see a sum over k, since it's \odot'ed with scalar q_{ij} times \tilde{M}j. Should this actually be w{i,j}? Please specify how this is done.
>
> As the reviewer pointed out, it should be w_{i,j}. We have corrected the equation. We thank the reviewer for pointing this out.

---

> > ### Author Response · Authors · 2022-11-15
> > **Response to reviwer LZWB part 2/2**
> >
> > >  it would be interesting to try time-domain based losses for this network, and see if that provides any improvement. Also, the architecture in the MixIT paper used mixture consistency, so that output sources sum up to the original input mixture. This might also be a useful constraint on the architecture here.
> >
> > We thank the reviewer for the thoughtful suggestions. As several papers on audio source separation show, it is indeed useful to consider phase and STFT consistency. In this work, we use real-valued masks (oracle binary masks for training) to directly compare with our base mode, SoP (Zhao et al., 2018). We will further explore the suggested ideas in future work.
> >
> > > I think best practice for reporting units in decibels is to use only one decimal place. Humans can often not even hear 0.1 dB of difference.
> >
> > We agree that humans can often not distinguish a 0.1 SDR difference. It seems that two decimal places are more often used in music separation community (SoP (Zhao et al., 2018), Music Demixing Challenge (Mistufuji et al.)) , while reporting one decimal place is common in other audio source separation tasks (including speech and universal separation). As many papers that use the MUSIC dataset, which we also use, report results with two decimal places, we followed the tradition.
> >
> > >More explanation of the motivation of NIT would be very welcome. My intuition is that it helps "soak up" extra noise by providing additional output sources, but this might not be right. Please add some explicit discussion of the motivation.
> >
> > Intuitively, noise tends to be less correlated to the image of the video than the sound from clearly visible objects. Thus, the strongly query-relevant sounds tend to be assigned to the query head. The noise heads indeed “soak up” less-correlated sounds. However, noise invariant heads tend to “soak up” too much as they can freely choose the source assignment that provides lower loss value. Thus, we use the proposed regularization. We provided the query vectors to the noise heads so that the noise heads can predict what will be extracted by the query heads and decide what to extract from the rest. We have clarified our motivation in Section 3.2.
> >
> > >Typos and minor comments: "For eaxmple," -> "For example,"
> >
> > We have corrected the typo. We thank the reviewer for pointing this out.

---

### Decision · Program_Chairs · 2023-01-20

**Decision:**

Accept: poster

**Justification For Why Not Higher Score:**

The paper builds on prior works for image-language and audio separation models, enabling a novel use case (text queried separation) and the ability to train on large scale data without the need for careful supervision. Although multiple reviewers highlighted novelty, there were also some concerns about certain parts of the model being simple extensions (like noise-invariant training). There are also prior works on text-queried audio separation.

**Justification For Why Not Lower Score:**

Novelty and evaluations were highlighted by all reviewers.

**Metareview: Summary, Strengths And Weaknesses:**

The authors present an algorithm for universal sound separation, to extract target audio from a mixture given text or image queries as input. Two systems are described, CLIPSep, which can be used to extract target sound using text queries, and CLIPSep-NIT, which additionally enables training on noisy (offscreen noise) audio. The reviewers agree that the work is novel, and enables training a text-driven separation model on noisy videos. Presented results are also thorough and convincing, along with links to demos.

**Note From Pc:**

if the above contains the word "oral" or "spotlight" please see: "oral" presentation means -> notable-top-5% and "spotlight" means -> notable-top-25%. As stated in our emails, we are disassociating presentation type from AC recommendations